

# Estimation of sea ice parameters from sea ice model with assimilated ice concentration and SST

Siva Prasad[1], Igor Zakharov[2], Peter McGuire[1,2], Desmond Power[2], and Richard Martin[3]

[1]Memorial University of Newfoundland, Canada
[2]C-CORE, Canada
[3]National Research Council of Canada

*Correspondence to:* Igor Zakharov (igor.zakharov@c-core.ca)

**Abstract.** A multi-category numerical sea ice model CICE along with data assimilation was used to derive sea ice parameters in the region of Baffin Bay and Labrador Sea. The assimilation of ice concentration was performed using the data derived from Advanced Microwave Scanning Radiometer (AMSR-E & AMSR2). The model uses a mixed layer slab ocean parametrization to compute the Sea Surface Temperature (SST) and thereby to compute the potential to freezing/melting of ice. The data from
Advanced Very High Resolution radiometer (AVHRR) was used to assimilate SST. The modeled ice parameters including concentration, ice thickness, freeboard, ridge height, and keel were compared with parameters estimated form remote sensing data. The ice thickness estimated from the model was compared with the measurements derived from Soil Moisture Ocean Salinity - Microwave Imaging Radiometer using Aperture Synthesis (SMOS-MIRAS). The ice concentration, thickness and freeboard estimates from model assimilated with both ice concentration and SST were found to be within the uncertainty
of the observation except during March. The model estimated draft was compared with the measurements from an upward looking sonar (ULS) deployed in the Labrador Sea (near Makkovik Bank). The difference between modeled draft and ULS measurements estimated from the model was found to be within 10 cm. The keel depth measurements from the ULS instruments were compared to the estimates from the model to retrieve a relationship between the ridge height and keel depth.

## 1 Introduction

Regional sea ice forecast is important for climate studies, operational activities including navigation, exploration of offshore mineral resources, and ecological applications, e.g. North Water polynya in Baffin Bay provides a warm environment for marine animals (Stirling, 1980).

Sea ice is a heterogeneous media and makes practically difficult for remote sensing instruments to measure the ice thickness, freeboard, and ridge parameters (Carsey, 1992). The climate forecast researchers depend on numerical modeling techniques
that implement the physical process of atmosphere and ocean on large scale computational platforms along with data assimilation methods to retrieve the information on sea ice parameters. Data assimilation methods can provide more accurate initial conditions for forecasting systems (Caya et al., 2006, 2010). The estimation of sea ice parameters is a challenging problem in the region of Baffin Bay and the Labrador Sea due to the high interannual variability of sea ice in this area (Fenty and Heimbach, 2013).



Previous sea ice modeling and assimilation studies at the Canadian Ice Service (CIS) (Sayed et al., 1999) provided an overview of an operational ice model coupled with atmospheric and ocean modules. The research (Sayed et al., 2001) compared the evolution of ice thickness distributions followed by the development of an operational ice dynamics model for CIS (Sayed et al., 2002). These modeling works were also improved by the data assimilation methods (Caya et al., 2006, 2010). The

Community Ice Ocean Model (CIOM) by Caya et al. (2006) used the Princeton Ocean Model for the simulation of ocean parameters and a multi-category ice model. The total ice fraction retrieved from the Special Sensor Microwave/Imager (SSM/I) was assimilated into CIOM model using 3D variational (3DVAR) technique (Caya et al., 2006) to estimate the ice concentration. The ice concentration estimates were further improved by assimilating information from both daily ice charts and RADARSAT (Caya et al., 2010). Assimilation studies by Lindsay (Lindsay, RW and Zhang, J, 2006) showed significant improvement in

assimilated ice concentration but with a large bias in the ice thickness pattern.

Cheng et al. (2012) presented a method for ice concentration and thickness analysis by combining the modeling of sea ice thermodynamics and the detection of ice motion by space-borne Synthetic Aperture Radar (SAR) data. The method showed promising results for sea ice concentration and ice thickness estimates. In another study, Ocean and Sea Ice Satellite Application Facility (OSI SAF) data were assimilated into Regional Ocean Modeling System (ROMS) for simulating sea ice

concentration and produced better results than the simulation without assimilation (Wang et al., 2013). Ice concentration and extent overestimated, probably due to the bias in atmospheric forcing, underestimation of heat flux and over/under estimation of sea ice growth/melt processes.

Sea ice models can be coupled to ocean and atmosphere models, but they can also be run in a standalone mode by prescribing the atmospheric and ocean conditions. The advantage of CICE model is the standalone capability. Here we use a combination of

modeling using the stand alone sea ice model, CICE (Hunke et al., 2015), the combination of optimal interpolation and nudging methods (Lindsay, RW and Zhang, J, 2006; Wang et al., 2013) to assimilate ice concentration. The optimal interpolation and nudging method is also used to assimilate SST. The optimal interpolation method is computationally cheap and was shown to provide better estimates than non-assimilated model (Wang et al., 2013). The simulated sea ice parameters are then validated with the observations in the region of the Baffin Bay and Labrador. This work uses high-resolution model configuration which

was previously described in the work of Prasad et al. (2015). The changes in ice concentration were taken into account to estimate the changes in the ice volume and thereby thickness estimates. In addition to validation of the ice concentration we discuss the effect of the assimilation on ice thickness, freeboard, draft and ridge keel. Since freeboard, draft and keel are functions of ice concentration and ice volume it is reasonable to compare the model values with corresponded observations. The work also estimates a relationship between ridge and keel using model values and ULS observation.

## 2   Model domain and forcing data

The sea ice model was implemented on a regional scale of about 10 km with a slab ocean mixed layer parameterization. Density-based criteria were used Prasad et al. (2015) to compute the mixed-layer depth and thereby compute the SST and the potential to grow or melt sea ice. The analysis of the non-assimilated model fo the sea ice concentration and its seasonal means



showed that the error associated with the model mostly spread across the area of the North Water polynya and the Davis Strait where the interaction of cold and warm water is frequent. In the present study, a data assimilation module is also introduced.

The surface atmospheric forcing is from high-resolution NARR data. The ocean forcing is from various sources: currents from CFSR, salinity from WOA-2013 and MLD computed from WOA-2013 Prasad et al. (2015). Atmospheric and ocean
forcing were used as inputs to the model. For SST, the climatology data derived from high-resolution NOAA were used as an input for the initial conditions and open boundaries. For the ice concentration and thickness, the initial condition is assumed as a no-ice state at the beginning of September. The assimilation starts from January 2005 and whenever remote sensing data are missing the model is not assimilated.

## 3    Remote Sensing Data for Assimilation and Validation

Several remote sensing data were used for the validation of ice parameters estimated by the model. Ice concentration derived from AMSRE of resolution $6 \times 4$ km Spreen et al. (2008) were used for validation of model estimated ice concentration. AMSRE was developed by JAXA, and it is deployed on Aqua satellite. AMSRE and AMSR2 are passive sensors that look at the emitted or reflected radiation from the earth's surface with multiple frequency bands. The vertical (V) and horizontal (H) polarization channels near 89 GHz were used to compute the ice concentration from AMSRE Spreen et al. (2008). The
Arctic Radiation and Turbulence Interaction STudy (ARTIST) sea ice algorithm used to determine ice concentration from AMSRE show excellent results above $65\%$ ice concentration where the error does not exceed $10\%$. With low ice concentrations, substantial deviations can occur depending on atmospheric conditions. The original AMSRE data with $6 \times 4$ km resolution scale were interpolated to the model grid before assimilating. The parameters of the sensor are provided in Table 1. AMSRE ice concentration were available from January 2005 to September 2011, after which it stopped functioning. From August 2012
AMSR2 had been used for data collection. The same frequency (89 GHz) as that of the AMSRE instrument was used to derive information from AMSR2. The spatial resolutions also remain the same for both AMSRE and AMSR2. The same algorithm was applied to derive ice concentrations from both AMSRE and AMSR2.

**Table 1.** Specifications of microwave radiometers used to estimate ice concentration.

| Specifications | AMSR-E/AMSR2 | SSMIS | |
|---|---|---|---|
| Center Frequency, GHz | 89 | 19 | 37 |
| Mean Spatial resolution, km | 5.4 | $69 \times 43$ | $37 \times 28$ |
| Polarization | HV | V | HV |
| Incidence angle, deg | 55 | 50 | |
| Swath, km | 1445 | 1700 | |
| Data availability, month/year | $08/2002 - 10/2011$ | $03/2005 -$Present | |

The assimilated model results of ice concentration were compared with the OSI SAF data. The details of the sensors are given in Table 1. OSI SAF product is derived from Special Sensor Microwave Imager Sounder (SSMIS) (Tonboe et al., 2016; Bell,



2006). The data is available on a 10 km polar stereographic grid and are derived from 19 V, 37 VH channels. The erroneous data were filtered out before the comparison. For SST assimilation of the measurements derived from AVHRR are used (Reynolds et al., 2007; Smith, 2016). SST data products are generated using a combination of satellite and in situ observations from buoy and ship observations and is available on a $0.25° \times 0.25°$ resolution.

CryoSat-2 altimeter operating in the SAR mode has the accuracy of about 1 cm with the spatial sampling about 45 cm (Bouzinac, 2014). The pulse limited footprint width in the across track direction is about 1.65 km and beam limited footprint width in the along-track direction is about 305 m (Scagliola, 2013), that corresponds to an along-track resolution about 401 m (assuming flat-Earth approximation). Therefore, the pulse-Doppler-limited footprint for SAR mode is about $0.6$ km$^2$. The CryoSat-2 freeboard and the ice-concentration products were generated at Alfred Wegener InstInstitute (AWI) (Ricker et al., 2014). The

products are available in a spherical Lambert azimuthal equal-area projection of 25 km resolution cell. The uncertainty of freeboard measurements can arise from speckle noise, lack of leads which causes the estimation of sea surface height unreliable, and snow cover. The uncertainty up to 40 cm can be observed in the region of Baffin Bay and Labrador Sea (Ricker et al., 2014).

For ice thickness, data of resolution 35 to 50 km derived from the SMOS Microwave Imaging Radiometer with Aperture

Synthesis (MIRAS) instrument (1.4-GHz channel) Kaleschke et al. (2012) is used. The ice thickness uncertainties are lower for thin ice and uncertainty increases as the thickness increases. SMOS data are available from 15 October 2010. The presence of snow accumulated over months also can increase the uncertainty. The uncertainty of SMOS ice thickness (observation) includes the error contributions, which are caused by the brightness temperature, ice temperature and ice salinity, see Table 2 Tian-Kunze et al. (2014); Ricker et al. (2016); Tietsche et al. (2017); Tietsche et al.. The insufficient knowledge on the

snow cover also introduces a large uncertainty in the thickness estimates from SMOS. Moreover, large errors occur during the melting period. In general, the uncertainty of thickness observation increases with increasing ice thickness, increasing snow cover and onset of melt Kaleschke et al. (2013). The SMOS ice thickness retrieval produces large uncertainty during the melt season and hence retrieval is not conducted during the melt season. Therefore, data from April to October are not available for our region of interest. Table 3 Kerr et al. (2001); Barré et al. (2008) shows the details on SMOS sensor.

**Table 2. SMOS sensor specifications.**

| Ice thickness | Uncertainty caused by a standard deviation of | | |
|---|---|---|---|
| | 0.5 K temperature brightness | 1 K ice tempearture | 1 g/Kg ice salinity |
| 0 -10 cm | < 1 cm | < 1cm | < 1cm |
| 10-30 cm | < 1 cm | 1-5 cm | 1- 13 cm |
| 30-50 cm | 1-4 cm | 2-10 cm | 2-22 cm |
| > 50 cm | > 4cm | > 7cm | ≤ 40 cm |

Ice draft measurements from ULS instrument located on the Makkovik Bank at $58.0652°$ W and $55.412°$ N (Ross et al., 2014), were used to analyze the ridge keel and the level ice draft in the region.



**Table 3.** SMOS sensor specifications.

| Polarization | HV |
|---|---|
| Incidence angle | $0-55°$ |
| Swath, km | 900 |
| Center Frequency (GHz) | 1.4 (L-band) |
| Mean Spatial resolution (km) | $30-50$ |
| Radiometric sensitivity over ocean, K | $2.5$ and $4.1$ |

The ULS data measured at an interval of approximately 5.5 seconds is available from the beginning of January to end of May during 2005, 2007 and 2009. The frequency histogram of the data yields a uni-modal, bi-modal or multi-modal distribution. A sample histogram is provided in Figure 1, for 10 February 2007. We assume that the first mode in the histogram corresponds to the level draft ice and the second mode corresponds to the ridge keel measurement. The first mode of the distribution is selected by finding a minimum between two peaks. The histogram was analyzed to derive daily averages of ice draft and keel measurements (Prasad et al., 2016).

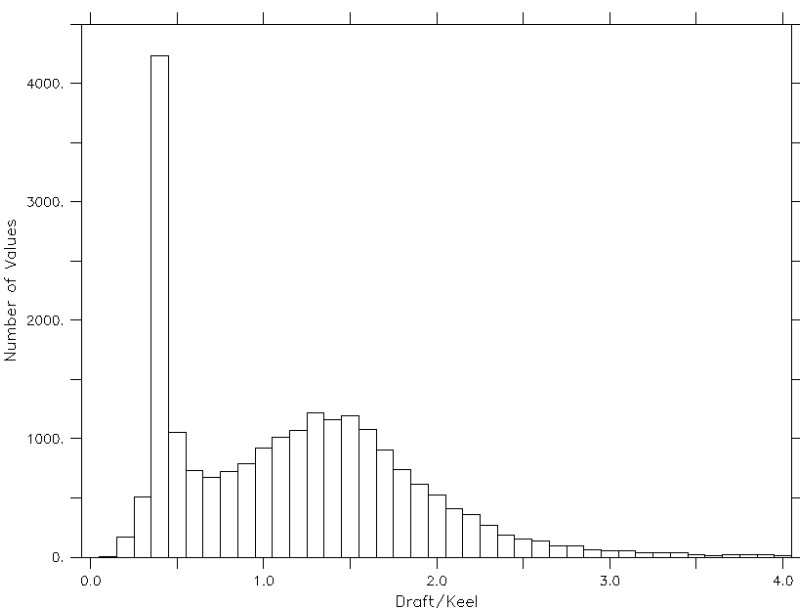

**Figure 1.** The histogram of the ULS measurement for the analysis of draft and keel (meters).





## 4  Data Assimilation

The assimilation module uses the combined optimal interpolation and nudging technique for ice concentration Lindsay, RW and Zhang, J (2006); Wang et al. (2013). The method can be represented generally as 1 Deutch (1965); Lindsay, RW and Zhang, J (2006).

$$X_a = X_b + dt \frac{K}{\tau}(X_o - X_b), \tag{1}$$

where $X_a$ is the final analysis of the variable, $X_o$ is the observed quantity (for ice concentration this is AMSR-E/AMSR2, for SST this is AVHRR), $X_b$ is the background estimate of the variable (for ice concentration and SST this is model estimate), $dt$ is the model time step, $\tau$ is the basic nudging time scale as in Wang et al. (2013), and $K$ is the nudging weight with the optimal interpolation value. K is computed as

$$K = \frac{\sigma_b^\alpha}{\sigma_b^\alpha + \sigma_o^2}, \tag{2}$$

where $\sigma_b$ and and $\sigma_o$ are the error standard deviation of the model estimate Deutch (1965) and the observations Deutch (1965) respectively The parameters in the weighing factor given in equation (2) is defined according to Lindsay, RW and Zhang, J (2006) as $\sigma_b = |X_o - X_b|$; $\sigma_o = 0.08$ (parameter may vary spatially), $\alpha = 6$.

When assimilation of ice concentration, $\sigma_o = 0.08$ is calculated from a long-term standard deviation to 0.08 since the AMSR-E/AMSR2 ice concentration error is unknown for values less than $65\%$. The parameter $\alpha = 6$, is used for the present study to ensure that the coefficients for assimilation are heavily weighted only when there is large variation between the model and the observation Lindsay, RW and Zhang, J (2006).

SST is also assimilated using the nudging and optimal interpolation scheme. For SST assimilation, $\sigma_o$ is fixed as $0.05$ to compensate for the assumption of zero mixed layer heat flux. A value $\alpha$ equal to 6 Lindsay, RW and Zhang, J (2006) was also used for the assimilation of SST to ensure that only large differences between the model and observation are weighted heavily

The assimilation of ice concentration is then followed by a re-computation of the estimated sea ice volume. The ice volume is subtracted or added by including the increments or decrements with specified ice thickness. Since a variable drag coefficient has been used for the friction associated with an effective sea ice surface roughness at the ice-atmosphere and ice-ocean interfaces and to compute the ice to ocean heat transfer the level ice area is updated by assuming the model deformed ice area and volume represents the realistic values.

## 5  Results and validation

Three model results are discussed here: model 'M0', the non-assimilated model; 'M1', the model assimilated with ice concentration from AMSR-E/AMSR2; and 'M2', the model assimilated with ice concentration from AMSR-E/AMSR2 and SST from



AVHRR. 'M2'assimilates only SST whenever there is a data gap in ice concentration from AMSR-E e.g. from 24 March 2005 to 31 March 2005 AMSR-E data are not available and, in that case, M2 assimilates only SST.

## 5.1 Ice concentration

Figure 2 column 1 shows the absolute error between the non-assimilated model and the OSI SAF, column 2 shows the absolute
5 error of the model assimilated only with ice concentration and OSI SAF, and column 3 shows the absolute error of the model assimilated with both ice concentration and SST and OSI SAF. Model M2 shows improvement in the ice concentration for January and March, but the results do not improve much

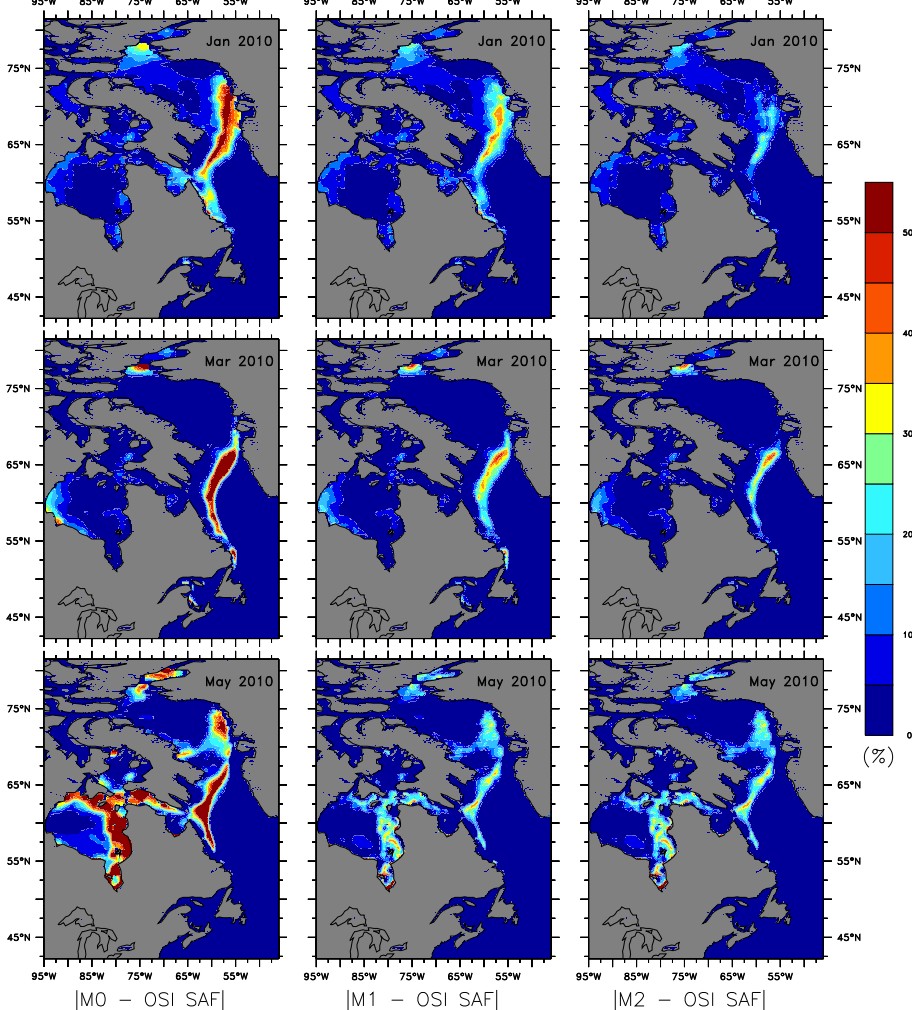

**Figure 2.** The absolute error of ice concentration from non-assimilated, assimilated models and OSI SAF for March 2010 and December 2010.



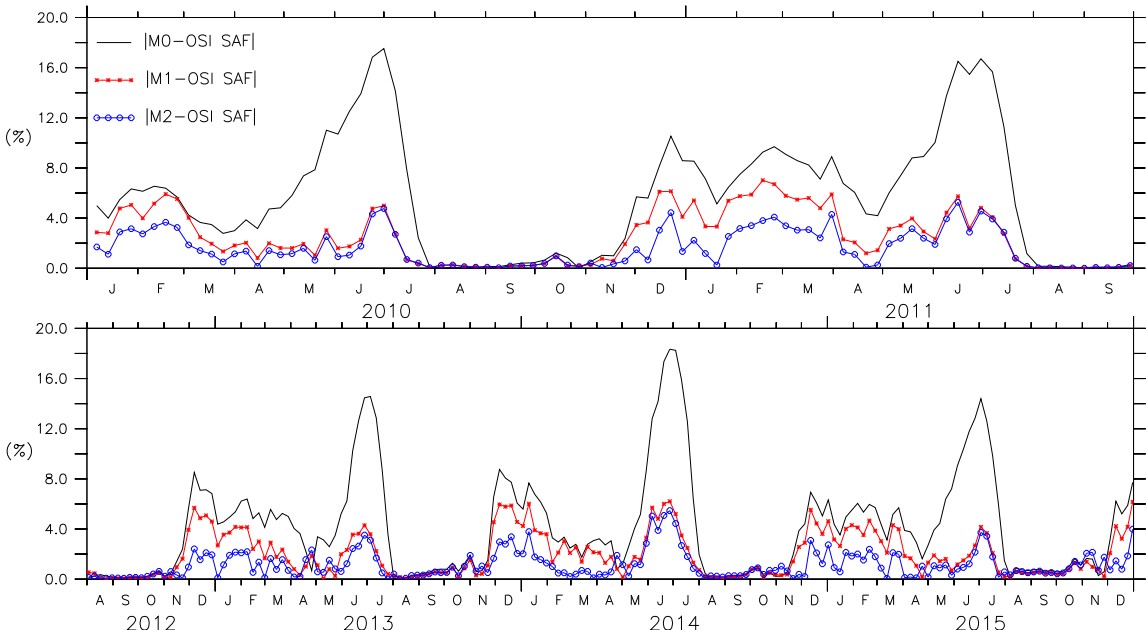

**Figure 3.** The absolute error for models M0, M1 and M2 from January 2010 to September 2011 is shown in row 1 and from August 2012 to December 2015 is shown in row 2.

Figure 3 shows the absolute error of the model and OSI SAF from January 2010 to September 2011 and the absolute error from August 2012 to December 2015. The assimilation of SST and ice concentration decreases the error between the model and the OSI SAF ice concentration. In 2010, the non-assimilated model error of $4.624\%$ was reduced to $1.939\%$ by assimilating ice concentration. The assimilation of SST and ice concentration decreased the error to about $1.118\%$ in 2010.

5      From October 2011 to July 2012, AMSR-E data are not available for a more extended period, and model M2 was assimilated only with SST, see Figure 4. During this period, the SST assimilation decreases the error between the model and the observation by almost $3\%$. The assimilation of ice concentration along with the assimilation of SST decreases the error in the ice concentration.

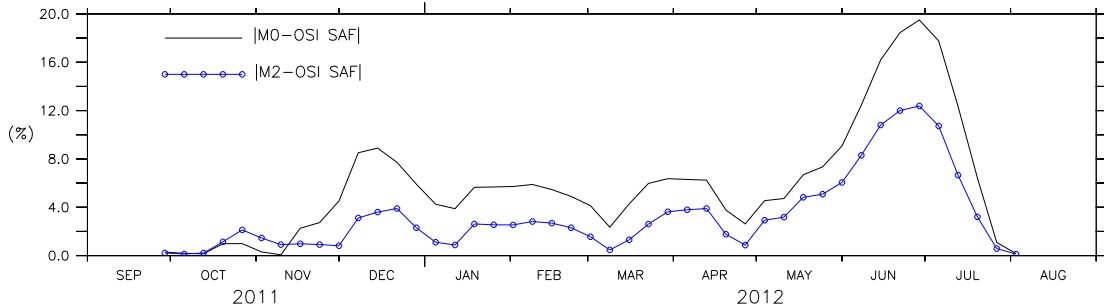

**Figure 4.** The absolute error from October 2011 to July 2012, ice concentration is not available for assimilation and hence model M2 will be only assimilated with SST during the period.





## 5.2 Ice thickness

In this section, we perform the comparison of ice thickness from the model with the observation. The large unacceptable uncertainties in observations create difficulties for the analysis. Also, it is strictly recommended not to use the SMOS data with an uncertainty greater than one meter (Tian-Kunze and Kaleschke, 2016) for practical applications. For comparison and

5    validation, ice thickness data from both the model and observation where the observed ice thickness has an uncertainty less than or equal to 100 cm are selected.The SMOS thickness has less uncertainty for thinner ice and higher uncertainty for thicker ice, see Table 2 for the uncertainty of SMOS ice thickness.

    Figures 5, 6, 7 shows the mean values of the thickness estimated from models M0, M1, M2 and SMOS with the uncertainty limits of the SMOS ice thickness (shaded gray). The values of Model M2 are within the uncertainty limits of SMOS ice thick-

10   ness until the end of February (except for 2014) end. In the case of SMOS derived thickness, the uncertainties would increase with the snow accumulation and melt onset. From the comparison, during March, the model results exceed the uncertainty limits. Compared with the uncertainty values, these results are in the acceptable range from October to the end of February. Figure 7 shows the results for the period October 2011 to April 2012 where AMSR-E data were missing during which M1 was not assimilated with ice concentration but used the initial conditions from the assimilated result. Model M2 used the initial con-

15   ditions assimilated with both ice concentration and SST but assimilates only SST during the period. Both models, M1 and M2, with the improved initial conditions show better forecasts in the long-term analysis. One of the reasons why the model values exceed the uncertainty limits during March is the choice of $\alpha = 6$, which considers only large differences while weighing the coefficient $K$. Since the assimilation shows improvement in ice thickness, using a value of $\alpha = 2$, it is expected to impose the model values within the uncertainty limits.

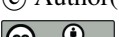



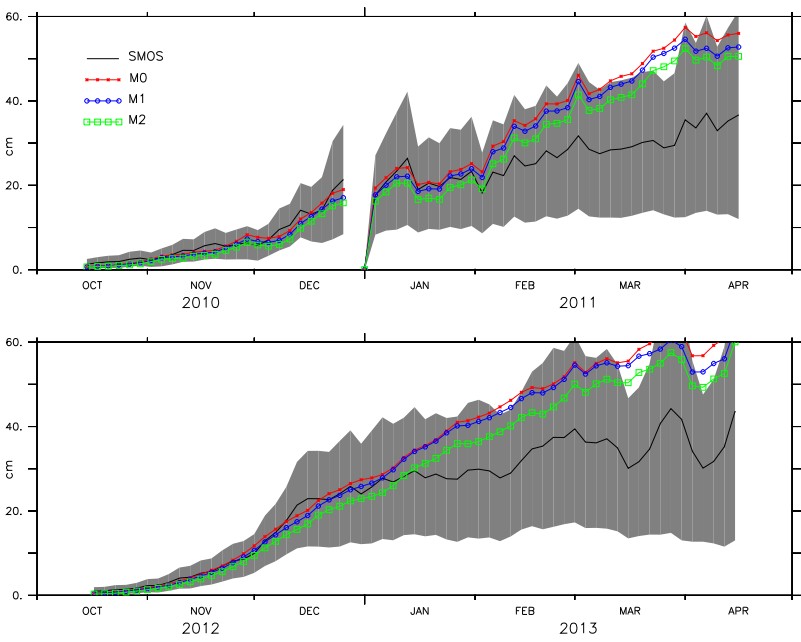

**Figure 5.** The ice thickness from the models M0, M1, M2, and observation (SMOS ice thickness) from October 2010 to April 2011 and October 2012 to April 2013 . The uncertainty of observation (SMOS ice thickness) is shaded in gray.

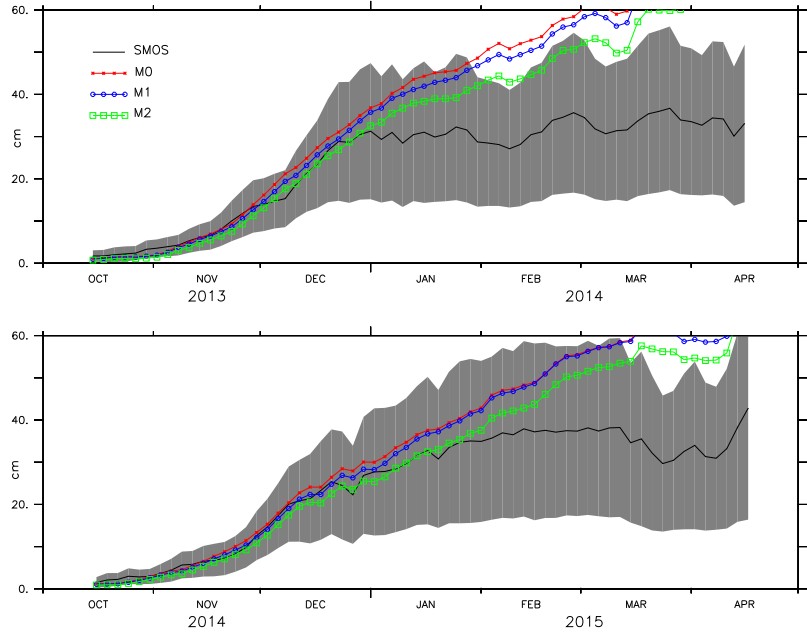

**Figure 6.** The ice thickness from the models M0, M1, M2, and observation (SMOS ice thickness) from October 2013 to April 2014 and October 2014 to April 2015. The uncertainty of observation (SMOS ice thickness) is shaded in gray.





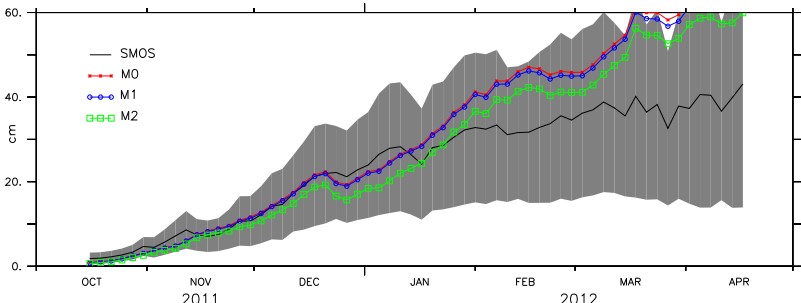

**Figure 7.** The ice thickness from models M0, M1(non assimilated but used the initial conditions from the model assimilated with ice concentration), M2 (assimilated only with SST and used model initial conditions derived from assimilating both ice concentration and SST) and observations (SMOS ice thickness) from October 2011 to April 2012. The uncertainty of observation (SMOS ice thickness) is shaded in gray.

The Model M2 thickness, SMOS derived ice thickness, and the uncertainty of the SMOS derived measurement for 15 December 2010, 15 January 2011 and 15 March 2011 is shown in Figure 8, and includes regions where observed uncertainties are larger than one meter.



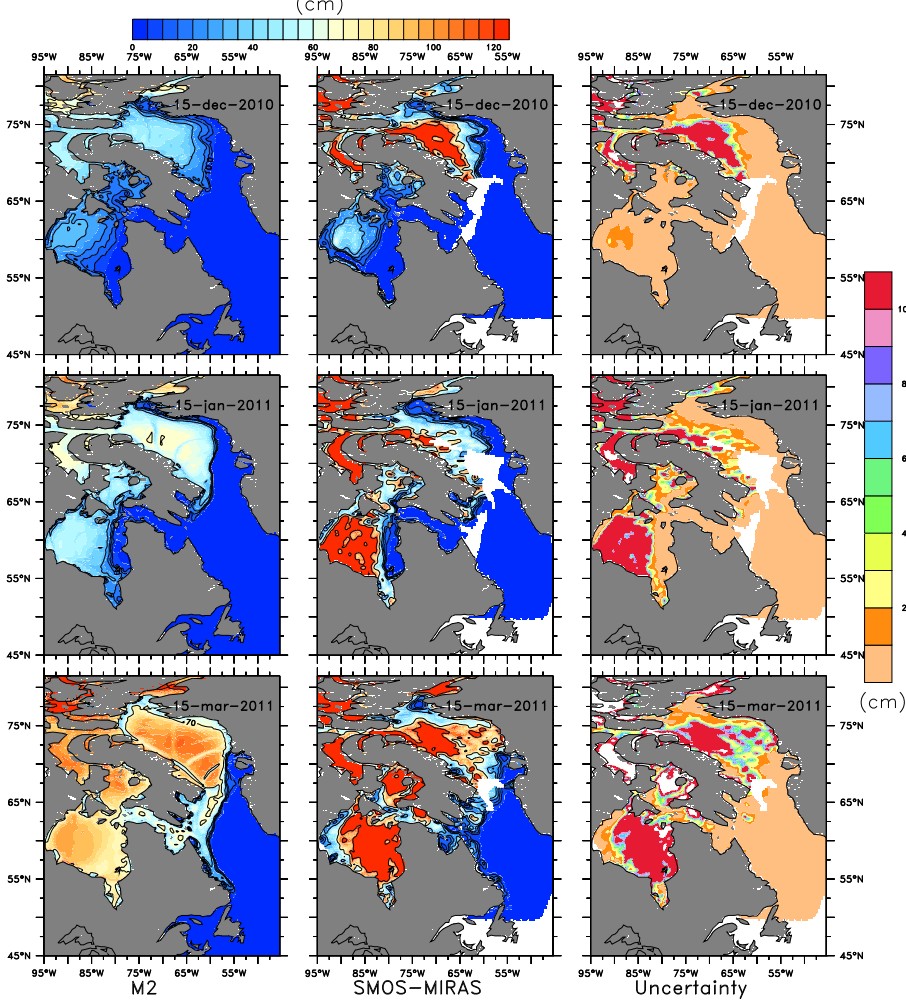

**Figure 8.** The model 'M2' estimated ice thickness, SMOS-MIRAS derived ice thickness, and the observation uncertainty for 2010-2011.

The thickness results for thin ice categories ($< 30cm$) from the model with SMOS are shown in Figures 9, 10, and 11. The thin ice category thicknesses are overestimated from October to November end but lies within the uncertainty limits of SMOS from December to March.



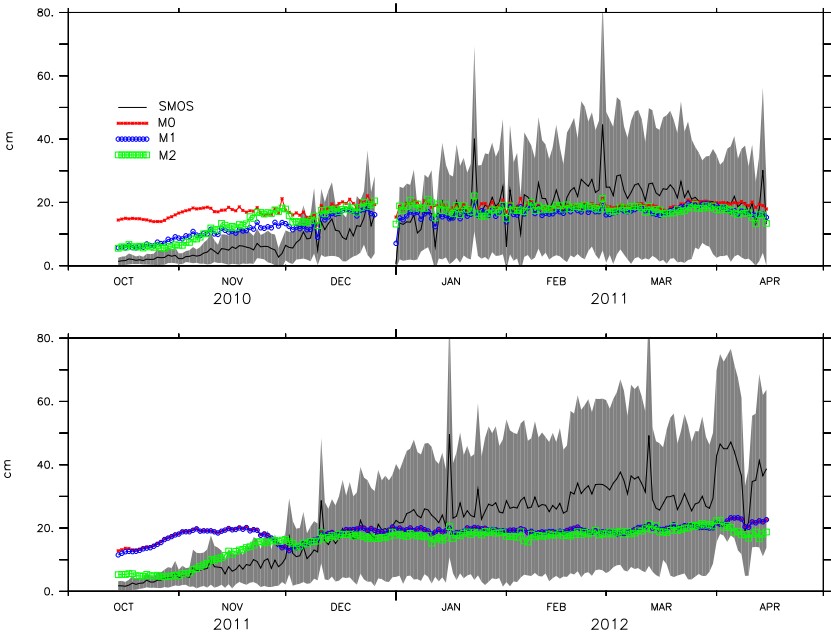

**Figure 9.** The model 'M2' estimated ice thickness, SMOS-MIRAS derived ice thickness, and the observation uncertainty for SMOS ice thickness less than 30 cm.

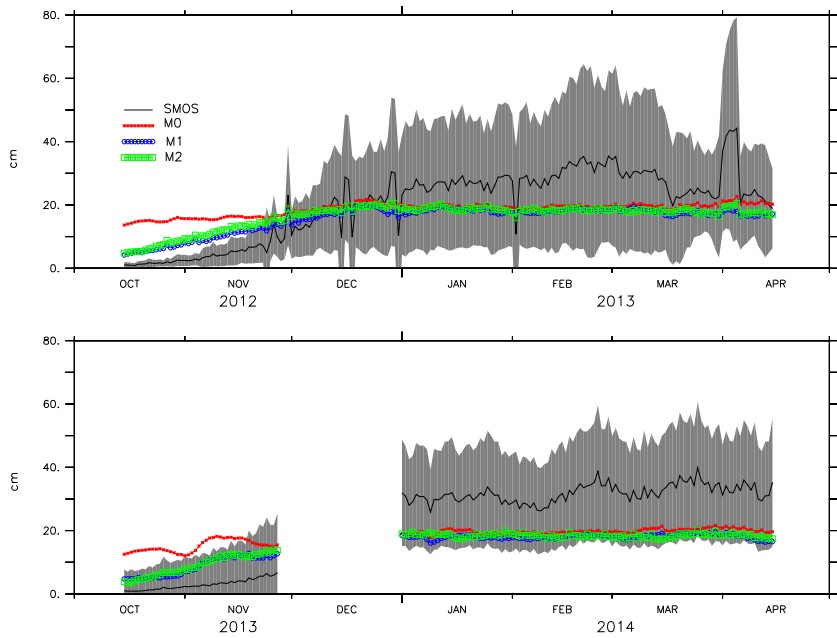

**Figure 10.** The model 'M2' estimated ice thickness, SMOS-MIRAS derived ice thickness, and the observation uncertainty for SMOS ice thickness less than 30 cm.



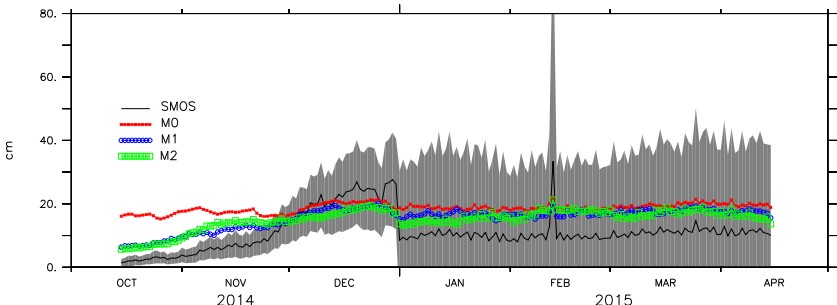

**Figure 11.** The model 'M2' estimated ice thickness, SMOS-MIRAS derived ice thickness, and the observation uncertainty for SMOS ice thickness less than 30 cm.

Figure 12, shows the SST from AVHRR with the shaded regions representing the observation uncertainty, SST from models M0, M1 and M2. The SST assimilation improves the ice concentration and ice thickness results for the model M2. The assimilated model M2 still has outliers observed during the winter period. This can be improved by decreasing choice of $\alpha$ (=6, presently) and by decreasing the nudging time scale (presently for SST nudging scale is 30 days). Decreasing the nudging

5  time scale can result in the late formation and early melt of ice (not shown here). The results can be improved with a choice of nudging time scale to be less frequent during the formation and more frequent during the winter till beginning or mid of March. Frequent nudging is also found to produce blow up for the thermodynamic model. Choice of the parameters in the assimilation has to be selected so that balance is maintained not to cause late formation and earlier melt and maintain the stability of the model thermodynamics and dynamics. For M0, non-assimilated model the results may be improved by including the mixed

10  layer heat flux with a parametrization similar to (Petty et al., 2014). Also, note that the model still assumes a fixed salinity profile and mixed layer profile.

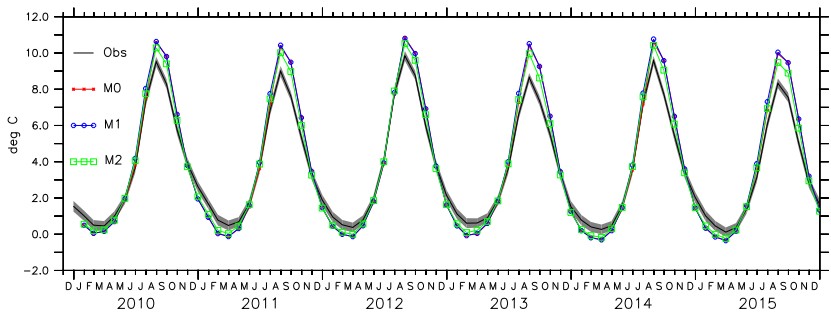

**Figure 12.** The SST from AVHRR with the shaded region represents the uncertainty of AVHRR SST, and SST from models M0, M1, M2.





### 5.3 Draft and keel depth

The ULS measurements were separated into level ice draft and keel depth measurement as described in Prasad et al. (2016)
The level ice draft, D is computed using equation (3) (Tsamados et al., 2014). The results are shown in Figure 13.

$$D = (\rho_i v_{ice} + \rho_s v_{sno})/(A\rho_w) \tag{3}$$

5        Where $\rho_i = 917 kg/m^3$ is the density of ice, $v_{ice}$ is the volume of ice, $\rho_s = 330.0 kg/m^3$ is the density of snow, $v_{sno}$ is the
volume of snow, A is ice concentration, $\rho_w = 1026 kg/m^3$.

Some deviations are noticed in the comparison of level ice draft. The estimated absolute error is about 10 cm for 2005, 2007,
2009. The error of 10cm can be accepted as a good correspondence between the ULS and Model. The discrepancy occurs due
to the fact that ULS gives values at a particular location with high resolution (within the footprint of several meters), while the
10      model is of 10 km resolution gives an averaged result close to the location of the ULS. Moreover, the analysis of histogram
from ULS shows multi-modal distribution at certain time points which indicates the presence of rafted ice. In the present study,
the rafted ice is also included and considered as the ridges which contribute towards the results achieved in this section.

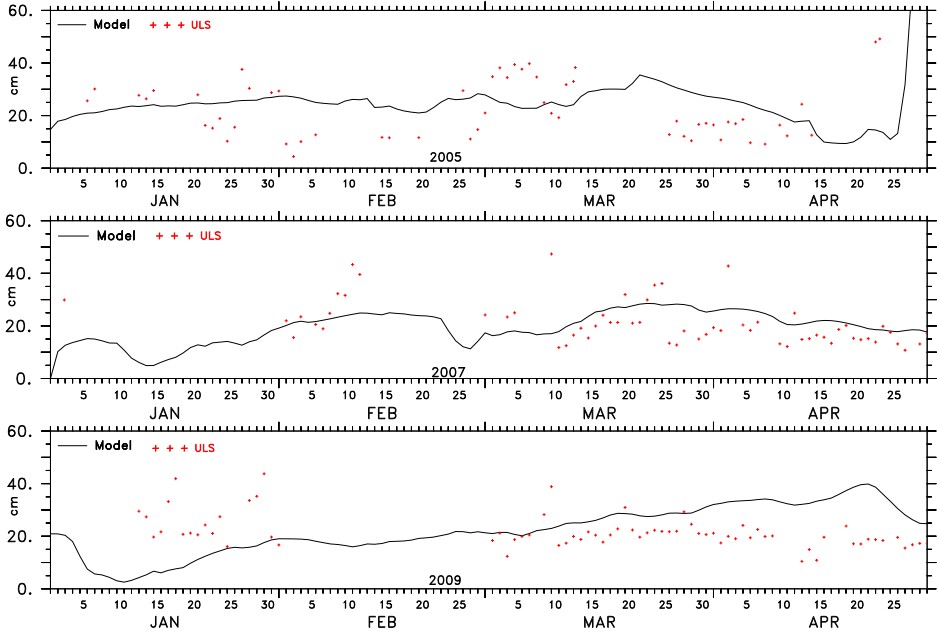

**Figure 13.** The level ice draft computed from the ULS measurement and the model estimated values in cm for 2005, 2007 and 2009.





The keel is computed using idealized sea ice floe comprising a system of two triangular sails and keels and a single melt pond (Tsamados et al., 2014). The ridge height is given by equation (4) and the correlation between the ridge height and keel depth is given by equation (5)

$$H_r = 2 \frac{V_{rdg}}{A_{rdg}} \frac{(\alpha D_k m_k + \beta C m_r)}{(\phi_r m_k D_k + \phi_k m_r C^2)} \tag{4}$$

Where $H_k$, is the keel depth, $H_r$ is the ridge height, $m_r = 0.4$ (21.8°) and $m_k = 0.5$ (26.5°) are the slopes of the ridge and the keel respectively, $\phi_r$ is the porosity of the ridges, $\phi_k = 0.14 + 0.73\phi_r$ (Shokr and Sinha, 2015) is the porosity of the keels. $D_k = 5$ is the ratio distance between ridge to distance between the keels. $V_{rdg}$ is the volume of the ridged ice, $A_{rdg}$ is the ridged ice area fraction, $\alpha$ and $\beta$ are the weight functions for area of ridged ice, $C$ is the coefficient that relates ridge to keel and

$$H_k = C H_r \tag{5}$$

gives the keel depth $H_k$. The Makkovik Bank where the keel measurements are estimated from ULS has high variability of ice thickness, and frequency of the formation of keels are high due to the combined effect of the Labrador currents and winds, rafted ice are common in this region. Here the model and the observation are used to estimate the parameter $C$.

The coefficient, $C$ estimated for 2005, 2007 and 2009 shows that a value between 3.00 and 4.50 gives a good estimate of
keel measurement for January and February while a value between 7.00 and 8.00 gives a good estimate for keel during March, April, and May. In Figure 14 the values of the coefficient $C$ that relates ridge to keel for January and February is 3 and $C = 7.00$ for March, April and May, see equation (5). These values are derived under the assumptions in equation (4). The sensitivity of parameters has to be further explored to determine the characteristics of each parameter and its effect on the ridge, keel relationship which may result in a different conclusion. Since the interest lies in deriving this relationship from the assimilated
model, so only results from M2 is presented. For non-assimilated model, the choice of parameters vary.

During January to February the formation of ice and ridges occurs, and during March the thick ice may be contributing towards the ridging thus increasing the value of $C$.

### 5.4    Freeboard

The uncertainty of freeboard measurements can arise due to the lack of leads. The presence of leads was ensured by selecting
the regions where lead fraction derived from CryoSat-2 was greater than zero. In the model, freeboard is computed using equation (6) (Tsamados et al., 2014). For the region, the uncertainty of the measurements is below 40 cm (Ricker et al., 2014).

$$D_f = (v_{ice} + v_{sno})/A - D \tag{6}$$

Where $v_{ice}$ is the volume of ice, $v_{sno}$ is the volume of snow, $A$ is the ice concentration, D is the draft, see equation (3).





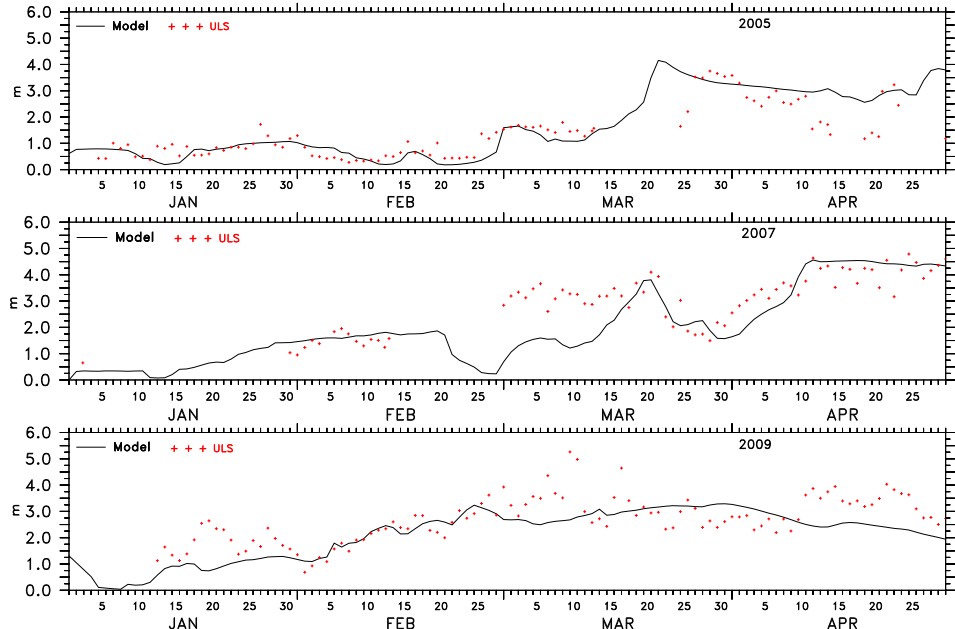

**Figure 14.** The keel depth computed from the ULS measurement and the model estimated values in cm for 2005, 2007 and 2009.

The absolute difference between the model and the observation for January, February and March 2011 is shown in the Figure 15. M2 freeboard measurements are close to the observed freeboard. Figure 16 demonstrates the spatial estimates with M2, observation and uncertainty.





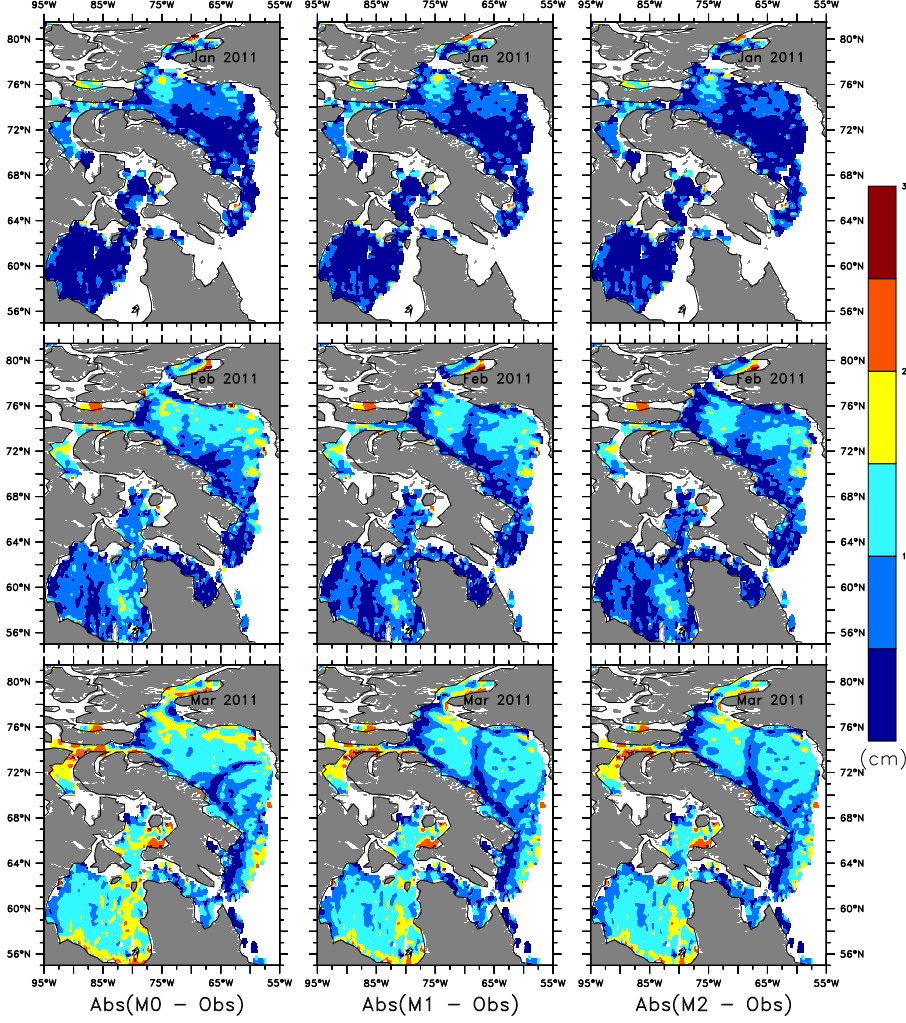

**Figure 15.** The absolute error between the model freeboard for M0, M1 and M2 and CryoSat-2 for January, February and March 2011.



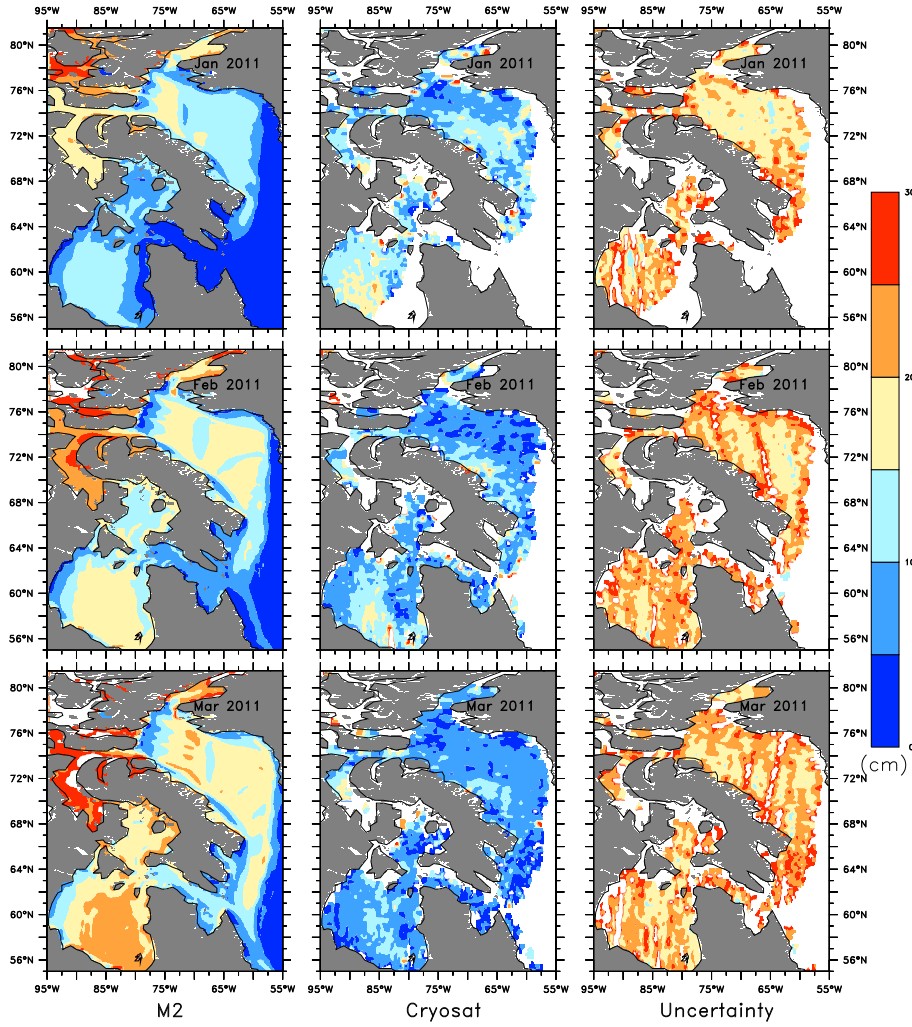

**Figure 16.** The Freeboard from model M2, CryoSat-2 and the uncertainty of the observations for 2011.

Figure 17 shows the observed freeboard from CryoSat-2, the uncertainty of observation, and the model M2. Only the model results from M2 are given since there are only slight deviations for M0 and M1 from the observation. Moreover, we are interested in the results of the assimilated model and how well it performs in the estimation of freeboard. The model values are within the uncertainty limits of the observation. Also, note that the model results are monthly averaged, while CryoSat-2 is 5 a mosaic of daily measurements within a month. The spatial average of freeboard for the region, the observed value, and the uncertainty is shown in Figure 16. The average freeboard from the model lies within the uncertainty limits of the observation.





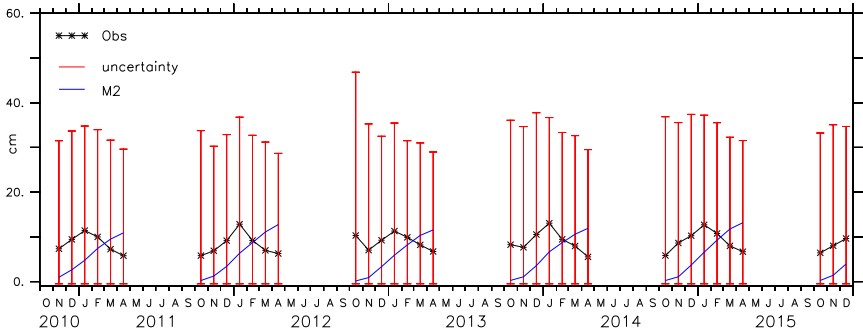

**Figure 17.** The freeboard from CryoSat-2, uncertainty of the observation and the model M2,

## 6   Conclusions

The assimilated models in the literature, and those implemented in forecasting centres use a constant drag formulation and lack the details on deriving the parameters other than ice concentration, and ice thickness. In this work a variable drag formulation is used for the friction associated with an effective sea ice surface roughness at the ice-atmosphere and ice-ocean interfaces and to

compute the ice to ocean heat transfer. The results from the updated model are compared with satellite derived measurements to validate the assimilation strategy. Moreover, the assimilated model results includes ice thickness, freeboard, and sail height, keel depth in addition to ice concentration.

The modeled ice thickness demonstrated a good correspondence with the estimates from SMOS-MIRAS, except during the period of maximum ice extent. The model freeboard are compared with estimates from CryoSat-2, and the RMSE was found to

range between $4.5$ cm and $11$ cm. The estimates of freeboard from the model are within the uncertainty values of the CryoSat-2 (below $40$ cm). The deviation in the results of ice thickness during March have to be further explored by tuning the parameters that contribute to the ice thickness in the non assimilated model as well as the assimilation parameters. The thin ice category thicknesses are overestimated from October to November end but lies within the uncertainty limits of SMOS from December to March. Also, the SMOS estimates are influenced by the presence of snow and also during the melt seasons the uncertainties

of SMOS estimated ice thickness might increase in which case comparison with more reliable data would be required.

The level ice draft and keel values derived from ULS were compared with the modeled values. The coefficient that related the sail height and keel depth for the Makkovick region lies in a range $3 - 8$ depending on the period of the year. The ULS data and model results were in agreement, except for some differences which can be explained by the difference in spatial resolution of the model and ULS data.

The assimilation methodology can be further improved by tuning the parameters, refining the error estimates derived from the observation data and combining data from several sources.

*Competing interests.*





*Acknowledgements.* Funding support was provided by Research and Development Corporation (RDC), Newfoundland and Labrador. The authors also thank Tony King (C-CORE) and Ingrid Peterson (DFO, Government of Canada) for providing ULS data from Makkovik Bank. We also thank Center for Health Informatics and Analytics (CHIA), MUN and ACENet, Canada for providing computational resources. We would like to than the developers of CICE, the Los Alamos sea ice model for public availability of the sea ice model and the users group.



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
