# Peer review of "Estimation of sea ice parameters from sea ice model with assimilated ice concentration and SST"

_The Cryosphere, 2018_

## Referee Comment (RC1) · Anonymous Referee #1 · 30 Jul 2018

Summary

This paper describes the use of a basic OI/nudging method to assimilate ice concentration and SST observations into an uncoupled version of the CICE model with a mixed-layer ocean parametrisation.

Unfortunately, this work is not currently up to the standard required for publication. A detailed review is given below, but the main reasons are as follows:

(1) It is unclear what is novel about this work. The conclusion states that the authors' use of a variable drag formulation is unique. However, Tsamados et al. (2014) previously incorporated a variable atmospheric and oceanic form drag into the CICE model. The paper has been cited by the authors, but they do not describe how their imple-

[Figure]

mentation of this method is related to any of their results. At the most basic level, the authors should show results with and without this formulation.

The statement that other centres do not provide details of parameters other than ice concentration or thickness is inaccurate, particularly as many of those centres are also using the same CICE model as the authors, with the same available parameters. The main conclusion of the results seems to be that assimilating observations brings the model output closer to reference observations, which is not a new result. Perhaps the differences in results when using thin ice of < 60 cm compared to < 30 cm might be an interesting angle, but this is not explored.

(2) The statements throughout the paper, that the model fits the validation observations well, are not backed up by the results themselves. Although the assimilation improves results, there remain clear systematic differences between the model output and the reference observations.

(3) There are many omissions in explanation of methods, several contradictions in the text, confusing wording, and the paper is missing references to current literature and relevant similar systems, e.g. TOPAZ, RIPS/RIOPS, ACNFS. Also missing is a description of how the authors' system differs and improves on these, and indeed what the purpose of the new system is. A number of the citations given are conference papers or otherwise unpublished works, which are not peer-reviewed and should not form a significant basis of citations.

General comments

The relevance of how this work fits in with the published literature needs to be discussed, along with other regional modelling systems. How do the results compare to e.g. coupled ocean-ice systems? What is the benefit of only using an SST parametrisation? Is this system to be used for operational or research purposes? A large number of the references used in this paper are unpublished or non-peer-reviewed works including conference papers. A more complete discussion of the peer-reviewed literature

is necessary.

The paper needs more information on all the input and validation datasets, including descriptions and data access information. The authors also need to ensure that all the datasets used are properly cited.

Why is the assimilation system set up to weight heavily in favour of the model rather than giving equal weight to the observations?

The paper repeatedly states that AVHRR data was assimilated. Actually, it looks like the authors are assimilating the AVHRR-only OISST analysis product, which although based on observations, is an analysis product. This needs to be made clear, along with information on the temporal resolution, timeliness etc of the product. Additionally, this product uses SSM/I and SSMIS information to create proxy SST observations for assimilation at high latitudes. This means the SST observations also include input from ice concentration data. Therefore, they are not independent from the SSM/I and SSMIS data being used for validation.

More detail and justification of which thickness ranges of SMOS and CryoSat-2 observations are being used is needed.

What real benefit is the assimilation giving? Figures 5-7 show that it brings the model closer to the observations, but it still deviates and all modelled ice thickness is too high. It is not convincing to state that the M2 model has good correspondence with the observations due to being in the uncertainty range, as even the free-running model is also managing that most of the time. Assimilating SST in addition to sea ice concentration produces better results, but few if any operational centres will not already be assimilating ice concentration and not SST observations.

The authors acknowledge that the assimilation is not optimised. If changing the value of alpha or adjusting the nudging timescale is expected to improve results, why has this not been done? Similarly for the relationship between ridge and keel.

The authors need to also include RMS (or standard deviation) statistics, as well as mean difference when discussing how well models match validation observations.

For figures 5,6,7 the modelled ice is too thick for all model runs after January. Although results for the M2 model are closer to observations than the M0 or M1 models, results are still not very good, which is not mentioned in the paper. In general throughout the paper, systematic biases or errors which are large in proportion to the model variable values are not addressed, or are dismissed as being within uncertainty levels. This shows a poor understanding of the validation results.

For results where the model thickness < 30 cm (figures 9,10,11), the models seem to be underestimating ice thickness rather than overestimating. This difference to the results seen in figures 5,6,7 needs to be addressed in the paper.

For figures 9,10,11 the modelled thin ice thickness remains roughly constant from December, and also the assimilation makes little difference. Reasons for this need to be addressed in the paper.

More information on methods is needed throughout. Instances of this are given under specific comments below.

The conclusions make a number of statements which are misleading. Details are given under specific comments below.

Many of the references are missing important information.

Specific comments

Line 6: Observations of ridge height and keel were not obtained from remote sensing data.

Line 8: What is your maximum SMOS thickness data? Only thin ice thickness available from SMOS.

Line 9: CryoSat-2 freeboard observations should be mentioned.

Line 15: Citations needed.

Line 19: A 1992 reference seems a strange choice given the recent advances in ice thickness remote sensing.

Line 20: Need to specifically relate this to sea ice forecasting.

Line 5-10: The relevance of this and how your work fits into this needs to be discussed, along with other regional modelling systems.

Line 11: Do you mean assessment rather than analysis here? Can only produce an analysis of ice concentration and thickness by assimilating ice concentration and thickness, not by modelling thermodynamics and (assimilating?) ice motion.

Line 12: which satellite?

Line 17: Before or after assimilation?

Line 19: Reference Hunke et al. (2015) the first time CICE is mentioned.

Line 22: Confusing, as you have mentioned prescribing ocean conditions but then mention you will be assimilating SST, before you mention the ocean mixed layer parametrisation below.

Line 31: "regional scale" - need to have a figure showing the domain.

Line 31: "about 10 km" - Need to mention what grid you are using.

Line 32: Should say "Density-based criteria were used >following< Prasad et al. (2015)..." and elsewhere, where the method has already been published.

Line 33: "analysis" should be "assessment" as the word analysis has a very specific meaning in the context of data assimilation.

Lines 3-6: Citations for the sources of all the forcing data are required.

Line 5: Why use SST climatology data rather than the daily analysis fields? What sort of climatology? Daily? Monthly?

Line 7: If assuming no ice at the start of the runs, important to state the spin-up time of the model (which should be mentioned anyway).

Line 10: Assimilating AMSR2/AMSR-E data, not using for validation.

Line 16: Mentioned that AMSR-E shows best results above 65% concentration, but are validating against SMOS observations of thin ice, as found in the MIZ where concentrations are much lower than 65%. Need to discuss limitations of the AMSR-E/AMSR2 data for the ranges relevant to the paper.

Line 18: AMSR-E data is interpolated to the model grid before assimilating (what about AMSR2?). The usual method would be to interpolate the model to the observation location. What is the benefit in interpolating the observations to the model grid?

Line 22: How consistent are observations derived from the different AMSR-E and AMSR2 instruments? Information needs to be added to the paper.

Table 1: Text says AMSR-E resolution is 6x4 km, but table says 5.4 km. Inconsistent. AMSR2 resolution is 5x3 km, so needs its own entry in the table. Additionally, not only SSMIS instruments in this time period - some were SSM/I (dates will depend on which OSI SAF product was used) so specifications for this instrument need to be included in the table as well.

Line 23: Which OSI SAF product number and version?

Line 24: Also uses SSM/I sensor.

Line 2: How were erroneous data removed? Methods needed.

Lines 2-4: Make it clear using AVHRR analysis, not SST measurements directly.

Line 5: CryoSat-2 altimeter is called SIRAL.

Line 12: Clarify what you are using the CryoSat-2 data for: validation. Why is the focus mainly on the SMOS data, and why are the CryoSat-2 SIRAL specifications not included in Table 1?

Line 14: Confusing that the SMOS thickness data resolution has a range. What is the resolution of the actual product used here? Also this is different to the range given in table 3.

Line 15-16, 19-20: Remove sentence "The ice thickness uncertainties are lower for thin ice and uncertainty increases as the ice thickness increases." as this is repeated below. Similarly for "Moreover, large errors occur during the melting period."

Line 16: Needs a line or two explaining how the SMOS sensor obtains measurements of ice thickness.

Line 20: What is the magnitude of the snow depth uncertainty?

Line 24: "...for our region of interest" - remove this, as not available in summertime for any Arctic region (and I don't think Antarctic SMOS ice thickness observations are available yet).

Line 24: Unclear what the Kerr and Barre citations are related to. Reword this.

Table 2: Caption is same as for table 3, update this.

Line 25: Show location of Makkovik Bank on map.

Table 3: Could this information be included in table 1? Are all these specifications directly relevant?

Line 2: Confusing - state which distribution is shown in the figure, and what causes the variation in distributions.

Line 3: Why only include data for a single day?

Line 4: What are these assumptions based on? Needs more explanation.

Line 6: What sort of quality control was undertaken for this data? Needs more explanation.

Figure 1: Needs units on x-axis, and date of observations in figure caption.

Line 7: For SST this is the AVHRR-only OISST analysis

Line 7: SST is not from model, it's a parametrisation

Line 11: "model estimate" should be "background model estimate" (as it's the background error in data assimilation terms)

Line 13: As sigma_o is different for sea ice concentration and SST and described below, remove from this line. Also "parameter may vary spatially" is confusing without additional explanation.

Line 15: If above 65% is 10%, this should be > 0.10 based on your stats given on page 3, line 16.

Line 16: I think this is intending to say something like "ensure that the assimilation is heavily weighted to the model background when there is large variation between the model and the observation." Needs rewording as it's unclear. However, method will weight towards model background even if the observation error is similar to the background. Why?

Line 22-25: Needs rewording as it's unclear what this means, and how this mechanism might directly affect the results.

Lines 1-2: Reword this as implies model assimilates SST instead of ice in data gaps. Also gap between AMSR-E and AMSR2 should be mentioned here. Need to state that the model is free-running during periods where no data is available for assimilation.

Line 4: "error" should be "mean difference", as the dataset being used as a reference is not necessarily "truth". This needs to be changed throughout the paper. Here, this should also say "absolute mean difference of ice concentration" for clarification. "OSI SAF" should be "OSI SAF dataset" (or similar wording).

Line 7: "the results do not improve much" Is this compared to Model M1? But in some locations the difference has reduced by about 20%, which is a good improvement. However, as you are assimilating the AVHRR-only OISST analysis, it is important to note that the product makes use of SSM/I and SSMIS ice concentration data to determine SST at high latitudes (though probably a different algorithm to the OSI SAF product). This means the SST observations you are assimilating are not truly independent from the SSM/I and SSMIS data you are using for validation. However, the AMSR-E/AMSR2 data is independent from the SSM/I and SSMIS data, and this should be stated.

Figure 3: Need to state that this is ice concentration and which product the models are being compared to in the figure caption. It also needs to be stated in the text somewhere what the spin-up period of the model is.

Lines 2-3: Need to state which instruments the assimilated ice concentration and OSI SAF data use.

Line 4: Why only giving the 2010 results? Also broken down into seasonal results would give a better picture.

Line 8: This last sentence does not relate to anything shown in figure 4, remove this or

improve discussion.

Line 2: Which model thickness category are you using for the comparison?

Line 2: observations from which instrument?

Lines 2-3: Unacceptable uncertainties in all observations? Confusingly worded.

Line 4: An uncertainty of 100 cm seems a lot for thin ice. What maximum ice thickness from SMOS are you using? From figures 5,6,7 it looks like 60 cm but this needs to be stated and explained. E.g. Xie et al. (2016; The Cryosphere, 10, 2745-2761) only use SMOS observations of < 40 cm, but others use up to 1 m thickness.

Line 5: How is model uncertainty determined?

Line 9: Add "As ice thickness increases through the season, so do the uncertainty limits."

Line 9: MO and M1 are too, except February 2013. What real benefit is the assimilation giving? Bringing closer to observations, but still deviates and are all still too high. Not convincing that it is only in the uncertainty range as even the free-running model is also managing that most of the time.

Line 10: Add "from October" before "until the end of February".

Lines 10-11: Move discussion of uncertainties to previous paragraph.

Line 12: Remove sentence beginning "Compared with the uncertainty values..." as this repeats information already stated.

Lines 16-19: This is because the assimilation is strongly weighted to model background. Demonstrates this is not the optimum set-up. If changing the value of alpha is expected to improve matters, why has this not been done?

Figures 5,6,7: Combine these into one figure. The correspondence with the observations is poor after about January. All modelled ice is too thick, and although results for the M2 model are closer to observations than the M0 or M1 models, results are still not very good. However, for results where the model thickness < 30 cm (figures 9,10,11) the models seem to be underestimating ice thickness rather than overestimating. This difference needs to be addressed in the paper.

Figure 7 caption: make clear that M1 is not assimilating ice concentration because there is no AMSR data available.

Lines 2-3: Why does figure 8 include regions where observed uncertainties are larger than 1 m, when on page 9 you have stated that this data has been rejected? This makes the figure very difficult to interpret, as it implies the model is underestimating ice thickness, but the comparisons in figures 5,6,7 indicate it is actually overestimating ice thickness for thin ice where the SMOS observations are more reliable - or underestimating for figure 9,10,11. Need to redo figure 8 showing only the relevant data, and also include panels with M2 differences to SMOS.

Figure 8 caption: only showing for 3 individual dates, not 2010-2011 - update caption to reflect this.

Figures 9,10,11: Maximum model thickness looks like 20 cm rather than 30 cm. Model underestimates thickness from December as thickness remains roughly constant throughout the year after this date. Also the assimilation makes little difference. Reasons for these results need to be addressed in the paper. Also caption states only model M2 is shown, but all models are shown on plot, update caption.

Figure 11: What is the cause of the discontinuity in SMOS ice thickness and uncertainty

between December and January? This needs to be addressed in the paper.

Figure 12: Needs to be larger, as it is difficult to see the shaded regions.

Line 1: How is the "observation uncertainty" generated? Is this actually the AVHRR-only OISST analysis uncertainty? Add this to text. This is not independent data as it's being assimilated for model M2. Could choose a different dataset for validation.

Line 2: Sentence beginning "The SST assimilation..." does not refer to figure 12. It is confusing to have this sentence here with no context.

Line 3: The model doesn't have "outliers", results show it has systematic biases in both summer and winter.

Lines 3-11: These lines give speculation on how the results could be improved, but this work needs to be done.

Line 2: Need to describe the method here, as Prasad et al. (2016) is a non-peer reviewed conference paper.

Line 6: Add that rho_w is the density of water.

Line 7: "about 10 cm" - give specific value (variation with season? Different years?). Need to add RMS or standard deviation.

Line 8: An error of 10 cm on a draft of 20 cm is proportionally very large, so can't be described as good correspondence.

Line 8: Reiterate here for benefit of reader that only done analysis for 2005, 2007, 2009 as this was when data was available.

Line 10: "close to the location of the ULS" - are you interpolating the model result to the observation location? If not what method is being used for the matchups and why?

Figure 13: State on figure caption where these measurements are located.

Lines 1-2: "single melt pond" - even in winter? This method needs more description.

Line 5: $H_k$ is not used in equation (4), remove (given below for equation (5)).

Line 6: $m_r$ and $m_k$ need more explanation - slopes given in degrees but what are 0.4 and 0.5?

Lines 5-8: Where are these values obtained from? Not all the variables have been given values either.

Line 12: Citation required for this statement.

Line 13: Model and observation of keel depth or ridge height? Confusing.

Line 16: Figure 14 only shows modelled and observed keel depth, not ridge height so can't see this relationship. Also need to give statistics for difference between modelled and observed keel depth.

Lines 17-19: If further work may result in a different conclusion, you need to do this further work to be able to draw any conclusions.

Line 25: How did you calculate lead fraction? Or cite existing product if that is what you used.

Line 26: Need to clarify in the text that the uncertainty given is for CS2 freeboard measurements. Need more information on the CS2 (CryoSat-2) product, e.g. how often available, where data was accessed etc.

Equation (6): I can't find this equation in Tsamados et al. (2014) but it looks like it's missing some brackets.

Line 1: "absolute difference" should be "absolute mean difference".

Line 2: "M2 freeboard measurements are close to the observed freeboard". This isn't true - figure 15 shows that the differences between the model runs and the observations are a large percentage of the actual values. There is also variation between the different months shown. It would be better to show differences rather than absolute differences on the spatial plot to be able to see where the biases are and in which direction. Other statistics such as RMS or standard deviation also need to be given.

Figure 16: Caption should specify Jan, Feb, March 2011 and not just 2011. Also need to show the difference plot and give other statistics e.g. RMS.

Line 4 (and line 6): The model values look systematically different to the observations in Figure 16. Figure 17 shows that the model is unable to replicate the seasonal changes in the freeboard observations, and just increases throughout the year.

Line 5: The data presumably still undergoes averaging if the points are observed multiple times within that month. Much more information on the dataset is needed.

Lines 2-3: Needs references to back this up.

Line 6: Misleading, as have not validated the assimilation method itself, only assimilated different combinations of observations.

Lines 6-7: This sentence implies the model is assimilating all these variables, which is incorrect. Reword.

Line 8: Disagree that it is a good correspondence.

Line 10: The RMSE should be mentioned previously with the rest of the results.

Line 11: Where have you split results into below 40 cm? Seems to be 60 cm for

observations or 30 cm for model category (which looks more like 20 cm).

Lines 14-15: No remote-sensing ice thickness data is available in the summertime due to the presence of melt ponds. As summertime data has been excluded in this study the error contribution from melt ponds is likely to be small.

Line 18: Agreement is not always close. This statement needs to be revised.

Lines 20-21: This needs to be done, rather than stating it as future work.

References:

Many of the references are missing important information, for example access URLs for technical reports, and format type e.g. book, report, dataset etc.

Technical corrections

There are a number of instances of citations not being in brackets when they should be, and vice versa. The authors need to go through the manuscript carefully to correct these.

The authors need to ensure that the paper is read by a native English speaker as there are a large number of minor but important grammatical errors. There are too many to list here.

There are also several areas where information is repeated within the same paragraph. Some are listed above. Correcting these instances would improve the readability of the paper.

---

## Referee Comment (RC2) · Anonymous Referee #2 · 30 Jul 2018

General Comments

In this paper, the authors assimilate ice concentration and AVHRR-derived SST into a 10 km CICE model for Baffin Bay and the Labrador Sea for the period spanning from 2010-2015. A series of 3 experiments are performed to assess the model's performance against ice thickness from SMOS, ice draft and keel depth from a ULS, and freeboard estimates versus CryoSat-2. A control run does not have any data assimilation, while the other two assimilate SST and SST and ice concentration. A nudging and optimal interpolation technique based on Lindsay and Zhang (2006) is used. Model mean ice thickness is compared against the SMOS ice thickness for the periods of Oct – March for the years 2010-2015. Overall, the "M2" test case which assimilates SST and ice concentration performs best, and is generally within the uncertainty bounds of

the SMOS data; however there is a significant positive bias shown for all years. An impressive comparison of the model's (M2) keel depth versus a ULS for 2005, 2007 and 2009 show very good agreement with data. However, model freeboard differences with CryoSat-2 data for Jan, Feb, and Mar 2011 show very little difference amongst the three test cases. Overall, while not "state-of-the-art", this paper shows some improvement with the assimilation of SST and ice concentration in a regional ice modeling system. I recommend publication with minor revisions.

Specific Comments

How are ice boundary conditions addressed in the model? Same technique as discussed in Prasad et al. 2015 paper? If yes, state this in the paper.

You use a 35-50 km SMOS ice thickness product for your thickness comparisons. You state that the SMOS data should not be used for thickness greater than 1 m; Figure 8 (middle column) shows a significant area of ice thicker than 1 m by March 15, 2011. Why didn't you consider using a merged CryoSat-2/SMOS ice thickness product such as is available from AWI? Do you have plans to assimilate ice thickness or freeboard into your model?

Page 1 line 19: why limit discussion to "climate forecast researchers"? This is important for operational sea ice modeling as well.

Page 3 lines 7-8: Why does the assimilation begin in January 2005? If the model is started from a no-ice state in September 2004, why doesn't assimilation begin in October 2004, when you should have data?

Page 3 line 11: Explain how you use AMSRE for validation of the model if you are assimilating that same data?

Page 4 line 1: What do you mean by "erroneous data"?

Page 7 lines 1-2: Why does M2 only assimilate SST when there are gaps in AMSR-E (and I assume AMSR2)? Why not assimilate all the time?

[Figure]

Page 8: Why is there no discussion on error reduction for the period from Nov 2012 – Dec 2015? A table of error stats would be helpful here.

Page 9: Have you tested different values of $\alpha$?

Page 15: How is snow measured or estimated in the ULS data? I assume the model results shown in Fig. 13 are for M2? If yes, state in figure caption. How do M0 and M1 compare here?

Page 17: I see little difference in Fig. 15 between column 2 and 3 in the plots. The paper states "M2 freeboard measurements are close to observed freeboard". I disagree. Perhaps the Jan 2011 looks best, but overall, the differences seem small for all 3 test cases.

Technical Corrections

Page 1 line 18: add "it" after "makes" and before "practically"

Page 2 line 7: rephrase to "into CDOM using a 3D"

Page 2 line 9: replace with "Lindsay and Zhang (2016)"

Page 2 line 16: "extent were overestimated"

Page 2 line 19: "of the CICE model"; which version of CICE is used? Specify in text

Page 2 line 20: ", and the combination"

Page 2 line 21: "(Lindsay and Zhang, 2006; Wang et al., 2013)"

Page 2 line 22: replace "cheap" with "inexpensive"

Page 2 line 24: "Baffin Bay and the Labrador Sea". "This work uses a high-resolution. . ."

Page 2 line 33: replace with "Density-based criteria (Prasad et al., 2015) to compute. . ."

Page 2 line 34: replace "fo" with "of"

Page 3 line 3: give reference for and spell out NARR

Page 3 lines 7-8: reword last sentence as it is confusing. Page 3 line 10: "remote sensing data sets"

Page 3 Table 1: add a column with dates for AMSR2

Page 3 last line: "The OSI SAF product"

Page 4 line 2-3: reword sentence to "Measurements derived from AVHRR (Reynolds et al., 2007; Smith, 2016) were used for SST assimilation."

Page 4 line 18: "ice thickness (observations) shown in Table 2 include..."

Page 4 line 19: "knowledge of the"

Page 4 line 22: "cover and the onset"

Page 4 line 25: "from an ULS"

Page 6: Reword first 2 sentences as they are poorly written

Page 5 line 12: here and in numerous places in the text, use "Lindsay and Zhang (2006)" as the reference, not the way it is written in the paper.

Page 7 lines 1-2: add parentheses to "AMSR-E (e.g., from 24 March 2005 to 31 March 2005), AMSR-E..."

Page 7 line 7: reword to "little improvement between M1 and M2 is shown for May 2010.

Figure 2: Makes dates Jan 2010, Mar 2010 and May 2010 BOLD black so they are easier to see. Caption is labeled wrong as it should be for Jan 2010, March 2010, and May 2010.

Page 3 Figure 3 legend: reword to "The absolute error for models M0, M1, and M2 from a) January 2010 to September 2011, and b) August 2012 to December 2015.

Figures 5-7: Make the uncertainty shading a lighter gray as it is difficult to see the other lines. What is the bias and RMSE for these comparisons?

Figure 8: same comment about making dates on plots readable. Can you add another column showing the difference between M2 and SMOS-MIRAS?

Figures 9-11: Make grey uncertainty lighter for readability. Captions not correct as M0 and M1 are shown (not just M2) as stated in Figure captions.

Page 14: Explain the difference in uncertainty shown in Fig 11 from Dec 31 to Jan 1.

Page 14 line 7: reword to "blow-ups" or something similar

Page 19: Make dates legible on all 9 plots.

Page 20 line 13: sentence with "November end but lies" does not make sense.

Page 22: provide a more complete reference for Deutch 1965

Page 23 line 21: Provide a date for first Tietsche reference

───────────────────

---

## Author Comment (AC1) · 24 Sep 2018

General Comments In this paper, the authors assimilate ice concentration and AVHRR-derived SST into a 10 km CICE model for Baffin Bay and the Labrador Sea for the period spanning from 2010-2015. A series of 3 experiments are performed to assess the model's performance against ice thickness from SMOS, ice draft and keel depth from a ULS, and freeboard estimates versus CryoSat-2. A control run does not have any data assimilation, while the other two assimilate SST and SST and ice concentration. A nudging and optimal interpolation technique based on Lindsay and Zhang (2006) is used. Model mean ice thickness is compared against the SMOS ice thickness for the periods of Oct – March for the years 2010-2015. Overall, the "M2" test case which assimilates SST and ice concentration

performs best, and is generally within the uncertainty bounds of the SMOS data; however there is a significant positive bias shown for all years. An impressive comparison of the model's (M2) keel depth versus a ULS for 2005, 2007 and 2009 show very good agreement with data. However, model freeboard differences with CryoSat-2 data for Jan, Feb, and Mar 2011 show very little difference amongst the three test cases. Overall, while not "state-of-the-art", this paper shows some improvement with the assimilation of SST and ice concentration in a regional ice modeling system. I recommend publication with minor revisions.

The authors would like to acknowledge the reviewer for the comments and suggestions.

Specific Comments How are ice boundary conditions addressed in the model? Same technique as discussed in Prasad et al. 2015 paper? If yes, state this in the paper.

The following text has been included for clarification "The net heat flux from the atmosphere is the upper boundary condition for ice thermodynamics. The heat flux from the ocean to the ice is the lower boundary condition. Based on temperature profile and boundary conditions the melt and growth of ice is computed. The open boundaries are configured in the same way as in (Hunke et al., 2010, Prasad et al., 2015)"

You use a 35-50 km SMOS ice thickness product for your thickness comparisons. You state that the SMOS data should not be used for thickness greater than 1 m; Figure 8 (middle column) shows a significant area of ice thicker than 1 m by March 15, 2011. Why didn't you consider using a merged CryoSat-2/SMOS ice thickness product such as is available from AWI? Do you have plans to assimilate ice thickness or freeboard into your model?

Figure 8 has been described in the following sentence "The Model M2 thickness, SMOS derived ice thickness, and the uncertainty of the SMOS derived measurement for 15 December 2010, 15 January 2011 and 15 March 2011 are shown in Figure 8, and include regions where observed uncertainties are larger than one meter" During the time the merged product was not available. We will use the merged product in the

future study. Yes, we do have plans to combine other products for assimilation.

Page 1 line 19: why limit discussion to "climate forecast researchers"? This is important for operational sea ice modeling as well.

The following text has been modified "The climate forecast researchers and operational ice modeling communities depend on numerical modeling techniques implementing the physical process of atmosphere and ocean on large scale computational platforms along with data assimilation methods to retrieve the information on sea ice parameters."

Page 3 lines 7-8: Why does the assimilation begin in January 2005? If the model is started from a no-ice state in September 2004, why doesn't assimilation begin in October 2004, when you should have data?

Please note that the AMSRE ice concentration product was available from January 2005 and hence assimilation started from the same period. Also, the model was given a 4 months spin-up.

Page 3 line 11: Explain how you use AMSRE for validation of the model if you are assimilating that same data? This was corrected AMSRE was used for assimilation and the product was compared with OSI SAF data. " Ice concentration derived from AMSRE of resolution 6 X 4 km (Spreen et al, 2008) were used for the assimilation of ice concentration."

Page 4 line 1: What do you mean by "erroneous data"? The following text has been modified for clarification. The erroneous data, were the ice concentration error was 100% or retrieval algorithm has failed were filtered out before the comparison.

Page 7 lines 1-2: Why does M2 only assimilate SST when there are gaps in AMSR-E (and I assume AMSR2)? Why not assimilate all the time? M2 assimilated SST only when ice concentration is not available for assimilation, otherwise the model assimilated both SST and ice concentration.

Page 8: Why is there no discussion on error reduction for the period from Nov 2012

– Dec 2015? A table of error stats would be helpful here. Only an example has been provided here. The rest of the results are shown as Figure 4.

Page 9: Have you tested different values of $\alpha$? Yes, different values of alpha were tested. A sensitivity of the parameter alpha has been shown in Lidsay et al. The value has to be further optimized considering the variable drag formulation variables for the model, which would be a future work.

Page 15: How is snow measured or estimated in the ULS data? I assume the model results shown in Fig. 13 are for M2? If yes, state in figure caption. How do M0 and M1 compare here? Upward looking sonar measures the draft from below and the measurement of snow is not available. Since we were interested in the results of assimilated model only M2 results are given.

Page 17: I see little difference in Fig. 15 between column 2 and 3 in the plots. The paper states "M2 freeboard measurements are close to observed freeboard". I disagree. Perhaps the Jan 2011 looks best, but overall, the differences seem small for all 3 test cases. Yes, these differences are very small. But M2 is found to be the best match with the observation.

Page 1 line 18: add "it" after "makes" and before "practically" Included

Page 2 line 7: rephrase to "into CDOM using a 3D" Rephrased

Page 2 line 9: replace with "Lindsay and Zhang (2016)" Replaced

Page 2 line 16: "extent were overestimated" Changed

Page 2 line 19: "of the CICE model"; which version of CICE is used? Specify in text Rephrased as "CICE version 5.1.2"

Page 2 line 20: ", and the combination" Rephrased

Page 2 line 21: "(Lindsay and Zhang, 2006; Wang et al., 2013)" Changed

[Figure]

Page 2 line 22: replace "cheap" with "inexpensive" Replaced

Page 2 line 24: "Baffin Bay and the Labrador Sea". resolution. . ." "This work uses a high-resolution..." Changed

Page 2 line 33: replace with "Density-based criteria (Prasad et al., 2015) to compute. . ." Rephrased as " Density-based criteria were used as in (Prasad et al., 2015) to compute the mixed-layer depth and thereby compute the SST and the potential to grow or melt sea ice."

---

## Author Comment (AC2) · 24 Sep 2018

Please note that the comments from referee are given in black font "Times new roman". Commnets from the authors are given in blue font "Calibri". The changes in the manuscript are given in green font "Calibri".

Summary This paper describes the use of a basic OI/nudging method to assimilate ice concentration and SST observations into an uncoupled version of the CICE model with a mixed-layer ocean parametrisation. Unfortunately, this work is not currently up to the standard required for publication. A detailed review is given below, but the main reasons are as follows:

[Figure]

(1) It is unclear what is novel about this work. The conclusion states that the authors' use of a variable drag formulation is unique. However, Tsamados et al. (2014) previously incorporated a variable atmospheric and oceanic form drag into the CICE model. The paper has been cited by the authors, but they do not describe how their implementation of this method is related to any of their results. At the most basic level, the authors should show results with and without this formulation. The statement that other centres do not provide details of parameters other than ice concentration or thickness is inaccurate, particularly as many of those centres are also using the same CICE model as the authors, with the same available parameters. The main conclusion of the results seems to be that assimilating observations brings the model output closer to reference observations, which is not a new result. Perhaps the differences in results when using thin ice of < 60 cm compared to < 30 cm might be an interesting angle, but this is not explored.

The authors would like to thank the reviewer for the comments and suggestions.

It is true Tsamados et al. (2014) has incorporated a variable atmospheric and oceanic form drag into the CICE model, but have not been used with a data assimilated model that produced parameters such as freeboard, sail, keel measurements. The forecast centers are using CICE model a constant variable drag formulation. Moreover, the used models are coupled with ocean models.

(2) The statements throughout the paper, that the model fits the validation observations well, are not backed up by the results themselves. Although the assimilation improves results, there remain clear systematic differences between the model output and the reference observations.

The study estimates the bias observed over years and will use it for further tuning of the model.

3) There are many omissions in explanation of methods, several contradictions in the text, confusing wording, and the paper is missing references to current literature and

relevant similar systems, e.g. TOPAZ, RIPS/RIOPS, ACNFS. Also missing is a description of how the authors' system differs and improves on these, and indeed what the purpose of the new system is. A number of the citations given are conference papers or otherwise unpublished works, which are not peer-reviewed and should not form a significant basis of citations.

The assimilation and tuning of the model is still an ongoing work and will be compared with RIPS and other models in the future. Moreover, the cited conference papers include the developments of RIPS. The novelty of the manuscript is the regional implementation of the uncoupled model.

General comments

The relevance of how this work fits in with the published literature needs to be discussed, along with other regional modelling systems. How do the results compare to e.g. coupled ocean-ice systems? The coupled ice-ocean systems are more complex and require additional tuning of ocean parameters. The purpose of the study was to go without an ocean model using a mixed layer parametrization and data assimilation to produce the best results. Coupling would require extensive work and is out of the scope of the current work.

What is the benefit of only using an SST parametrization? Is this system to be used for operational or research purposes? A large number of the references used in this paper are unpublished or non-peer-reviewed works including conference papers. A more complete discussion of the peer-reviewed literature Discussion paper is necessary.

Please note that the SST parametrization was discussed earlier in Parasad et al. (2015). Similar to RIPS 2016 the model used the same density parametrization but with a different criteria for forwarding the slab ocean model parametrization.

The paper needs more information on all the input and validation data sets, including descriptions and data access information. The authors also need to ensure that all

the datasets used are properly cited. All the input and validation data sets had been described and properly cited.

Why is the assimilation system set up to weight heavily in favour of the model rather than giving equal weight to the observations?

The assimilation followed work of Lindsay et al. Further tuning of the model physics and assimilation schemes have to be done to better understand the model behavior. The system is actually weighted heavily in marginal ice zones where significant bias can occur.

The paper repeatedly states that AVHRR data was assimilated. Actually, it looks like the authors are assimilating the AVHRR-only OISST analysis product, which although based on observations, is an analysis product. This needs to be made clear, along with information on the temporal resolution, timeliness etc of the product. Additionally, this product uses SSM/I and SSMIS information to create proxy SST observations for assimilation at high latitudes. This means the SST observations also include input from ice concentration data. Therefore, they are not independent from the SSM/I and SSMIS data being used for validation.

Yes, SST used an analysis product since model is assimilated over its domain including at ice edges where ice concentration has lower values. It was clarified in the paper that AVHRR-only OISST analysis product uses ice information to retrieve SST only for regions where ice concentration is greater than 0.5. The following sentence has been included for clarity " The analysis product estimates SST from ice concentration only in regions where ice concentration is greater than 50%, otherwise uses satellite data to retrieve SST values."

More detail and justification of which thickness ranges of SMOS and CryoSat-2 observations are being used is needed. The justification has been provided in the paper "Also, it is strictly recommended not to use the SMOS data with an uncertainty greater than one meter (Tian-Kunze and Kaleschke, 2016) for practical applications." This has

been clearly stated in the user manual of the SMOS product. For Cryostat-2 freeboard measures had been used for comparison. Since thickness estimates of CryoSat 2 is derived from freeboard estimates we think that it would be best to compare freeboad estimates with Cryosat-2 instead of thickness estimates.

What real benefit is the assimilation giving? Figures 5-7 show that it brings the model closer to the observations, but it still deviates and all modelled ice thickness is too high. It is not convincing to state that the M2 model has good correspondence with the observations due to being in the uncertainty range, as even the free-running model is also managing that most of the time. Assimilating SST in addition to sea ice concentration produces better results, but few if any operational centres will not already be assimilating ice concentration and not SST observations.

All modeling centres assimilate SST in Ocean model and passe the information to sea ice model. Here, we state that if we use analysis product of SST in the ice model itself it produces better results. Moreover, operational centers give information on ice concentration but ice thickness significant biases are observed also, other information such as ridge height, keel depth, freeboard data are rarely discussed.

The authors acknowledge that the assimilation is not optimized. If changing the value of alpha or adjusting the nudging timescale is expected to improve results, why has this not been done? Similarly for the relationship between ridge and keel.

This is a parameter sensitivity study and is an ongoing work.

The authors need to also include RMS (or standard deviation) statistics, as well as mean difference when discussing how well models match validation observations. For figures 5,6,7 the modelled ice is too thick for all model runs after January. Although results for the M2 model are closer to observations than the M0 or M1 models, results are still not very good, which is not mentioned in the paper. In general throughout the paper, systematic biases or errors which are large in proportion to the model variable values are not addressed, or are dismissed as being within uncertainty levels. This

shows a poor understanding of the validation results.

For results where the model thickness < 30 cm (figures 9,10,11), the models seem to be underestimating ice thickness rather than overestimating. This difference to the results seen in figures 5,6,7 needs to be addressed in the paper.

For figures 9,10,11 the modelled thin ice thickness remains roughly constant from December, and also the assimilation makes little difference. Reasons for this need to be addressed in the paper.

More information on methods is needed throughout. Instances of this are given under specific comments below.

The conclusions make a number of statements which are misleading. Details are given under specific comments below.

Many of the references are missing important information.

Specific Comments

Page 1 Line 6: Observations of ridge height and keel were not obtained from remote sensing data

The ridge height is not estimated from remote sensing data but keel depth is estimated using the Upward Looking Sonar Instrument. The following text was edited in the Manuscript "The modeled ice parameters including concentration, ice thickness, freeboard, and keel depth were compared with parameters estimated form remote sensing data."

Line 8: What is your maximum SMOS thickness data? Only thin ice thickness available from SMOS. The maximum SMOS thickness is not mentioned. Instead we consider the thickness levels that are used for practical applications. Please note that for practical applications ice thicknesses with uncertainty greater than 1m is not used. In several experiments we used SMOS values below 50cm.

Line 9: CryoSat-2 freeboard observations should be mentioned. Included the following text "The model freeboard estimates were compared with the freeboard measurements derived from CryoSat2."

Line 15: Citations needed. This is general introduction and the readers know about importance of sea ice forecast.

Line 19: A 1992 reference seems a strange choice given the recent advances in ice thickness remote sensing. The heterogeneous property of sea ice still poses a challenge for measuring several sea ice parameters.

Line 20: Need to specifically relate this to sea ice forecasting. Please see the next sentence "Data assimilation methods can provide more accurate initial conditions for forecasting systems"

Page 2 Line 5-10: The relevance of this and how your work fits into this needs to be discussed, along with other regional modelling systems. In the sea ice data assimilation literature a detailed description of the parameters other than ice thickness is not provided. Even the discussions that already exists tend to omit the uncertainty of the ice thickness.

The following text was edited "In addition to validation of the ice concentration we discuss the effect of the assimilation on ice thickness, freeboard, draft and ridge keel. Since freeboard, draft and keel are functions of ice concentration and ice volume it is reasonable to compare the model values with corresponded observations. The work suggests a methodology to extract the level ice draft and keel depth information from ULS measurements, which was then used to describe the relationship between ridge and keel."

Line 11: Do you mean assessment rather than analysis here? Can only produce an analysis of ice concentration and thickness by assimilating ice concentration and thickness, not by modelling thermodynamics and (assimilating?) ice motion.

Yes, we mean assessment. This was corrected in the manuscript.

Line 12: which satellite? RADARSAT-1 and RADARSAT-2

Line 17: Before or after assimilation? After assimilation Following line was edited "Ice concentration and extent was overestimated in the assimilated model, probably due to the bias in atmospheric forcing, underestimation of heat flux and over/under estimation of sea ice growth/melt processes."

Line 19: Reference Hunke et al. (2015) the first time CICE is mentioned. Added.

Line 22: Confusing, as you have mentioned prescribing ocean conditions but then mention you will be assimilating SST, before you mention the ocean mixed layer parametrisation below. The following sentence was edited for clarification.

"The optimal interpolation and nudging method is also used to assimilate SST estimated by a slab ocean parametrization in the sea ice model."

Line 31: "regional scale" - need to have a figure showing the domain.

Line 31: "about 10 km" - Need to mention what grid you are using. The following text was edited for clarification "The sea ice model was implemented on a regional scale of about 10 km orthogonal curvilinear grid with a slab ocean mixed layer parameterization."

Line 32: Should say "Density-based criteria were used >following< Prasad et al. (2015)..." and elsewhere, where the method has already been published. Included

Line 33: "analysis" should be "assessment" as the word analysis has a very specific meaning in the context of data assimilation. Corrected

Lines 3-6: Citations for the sources of all the forcing data are required.

Line 5: Why use SST climatology data rather than the daily analysis fields? What sort

of climatology? Daily? Monthly? Monthly climatology. The sentence has been revised to clarify this.

"For SST, the climatology data derived from high-resolution NOAA were used as an input for the initial and boundary conditions"

Line 7: If assuming no ice at the start of the runs, important to state the spin-up time of the model (which should be mentioned anyway). The model is spin-up for 4 months before assimilation. Please note that the experiments showed that 10 year spin up also produced similar result for free run.

Line 10: Assimilating AMSR2/AMSR-E data, not using for validation. Revised

Line 16: Mentioned that AMSR-E shows best results above 65% concentration, but are validating against SMOS observations of thin ice, as found in the MIZ where concentrations are much lower than 65%. Need to discuss limitations of the AMSR-E/AMSR2 data for the ranges relevant to the paper.

We do not consider this in the present case since we use a long term standard deviation to assimilated AMSR-E data.

Line 18: AMSR-E data is interpolated to the model grid before assimilating (what about AMSR2?). The usual method would be to interpolate the model to the observation location. What is the benefit in interpolating the observations to the model grid?

Yes, the data assimilation methods do have an operator to observational space in our case it would be an interpolation and after assimilation it is interpolated back to the model grid. We currently neglect this forward operator. Interpolating it to model grid allows computational benefit, easiness of implementation. Moreover, since we introduce constant values for sigma, the observations are considered to have a constant uncertainty value and hence we assume the error would be less if the observation is interpolated to the model grid. Although it was also evident from the results it would be presented as a separate work later.

Line 22: How consistent are observations derived from the different AMSR-E and AMSR2 instruments? Information needs to be added to the paper. This was already stated in the section on "Remote Sensing Data for Assimilation and Validation" "The same frequency (89 GHz) as in the AMSRE instrument was used to derive information from AMSR2. The spatial resolutions also remain the same for both AMSRE and AMSR2. The same algorithm was applied to derive ice concentrations from both AMSRE and AMSR2."

Table 1: Text says AMSR-E resolution is 6x4 km, but table says 5.4 km. Inconsistent. AMSR2 resolution is 5x3 km, so needs its own entry in the table. Additionally, not only SSMIS instruments in this time period - some were SSM/I (dates will depend on which OSI SAF product was used) so specifications for this instrument need to be included in the table as well

Please note that in the table mean spatial resolution was provided. The table has been updated. The product version 1.4 had been used and the documentation indicates SSMIS satellite data.

Line 23: Which OSI SAF product number and version? Product OSI-401b and version 1.4

Line 24: Also uses SSM/I sensor. Yes.

Page 4 Line 2: How were erroneous data removed? Methods needed. Erroneous data are indicated by flags so these data that carry the flags are removed before interpolating to model grid.

Lines 2-4: Make it clear using AVHRR analysis, not SST measurements directly. Revised.

Line 5: CryoSat-2 altimeter is called SIRAL. Added

Line 12: Clarify what you are using the CryoSat-2 data for: validation. Why is the focus mainly on the SMOS data, and why are the CryoSat-2 SIRAL specifications not

included in Table 1?

Revised as "Freeboard measurements from CryoSat-2 altimeter were used to compare the freeboard estimates by the model."

Line 14: Confusing that the SMOS thickness data resolution has a range. What is the resolution of the actual product used here? Also this is different to the range given in table 3.

Table 3 is corrected. The actual product resolution used here is 12.5km x 12.5km The following text has been edited to clarify

"For ice thickness, data product derived from the SMOS Microwave Imaging Radiometer with Aperture Synthesis (MIRAS) instrument (1.4-GHz channel) (Kaleschke 2012) on a grid resolution of 12.5 x 12.5km is used."

Line 15-16, 19-20: Remove sentence "The ice thickness uncertainties are lower for thin ice and uncertainty increases as the ice thickness increases." as this is repeated below. Similarly for "Moreover, large errors occur during the melting period." Removed.

Line 16: Needs a line or two explaining how the SMOS sensor obtains measurements of ice thickness. The following sentence is included "The ice thickness is retrieved from observation of the L-band microwave sensor of SMOS. Horizontal and vertical polarized brightness temperatures in the incidence range of < 40 degree are averaged. The ice thickness is then inferred from a three layer (ocean-ice-atmosphere) dielectric slab model. "

Line 20: What is the magnitude of the snow depth uncertainty? Following text has been included for clarification "The insufficient knowledge of the snow cover also introduces a large uncertainty in ice thickness estimates. Snow depth uncertainity can be 50-70% of mean value (zhou et al., 2018)."

Line 24: "...for our region of interest" - remove this, as not available in summertime for any Arctic region (and I don't think Antarctic SMOS ice thickness observations are

available yet). Removed

Line 24: Unclear what the Kerr and Barre citations are related to. Reword this. Reworded

Table 2: Caption is same as for table 3, update this. Caption is updated "SMOS uncertainty"

Line 25: Show location of Makkovik Bank on map. Shown as Figure 1.

Page 5 Table 3: Could this information be included in table 1? Are all these specifications directly relevant? No this is information about sensor specification while table 1 shows the uncertainties of SMOS data.

Line 2: Confusing - state which distribution is shown in the figure, and what causes the variation in distributions. The modes in the figure are clarified in the following sentence

"We assume that the first mode in the histogram corresponds to the level draft ice and the second mode corresponds to the keel depth measurement." The sea ice has level and deformed parts and moreover the ice is in constant motion. The deformation variations are captured in the second mode while the non deformed part is captured in the first mode.

Line 3: Why only include data for a single day? This is only a sample. Including data for every day is not possible as data is available for 2005, 2007, 2009 form January till April/May. Moreover, we are performing ULS analysis for each day.

Line 4: What are these assumptions based on? Needs more explanation. The modes in the figure are clarified in the following sentence

"We assume that the first mode in the histogram corresponds to the level draft ice and the second mode corresponds to the keel depth measurement."

The sea ice has level and deformed parts and moreover the ice is in constant motion. The deformation variations are captured in the second mode while the non deformed

part is captured in the first mode.

Line 6: What sort of quality control was undertaken for this data? Needs more explanation. The data is available after the quality control and hence not discussed here.

Figure 1: Needs units on x-axis, and date of observations in figure caption. Mentioned meters in X

Page 6 Line 7: For SST this is the AVHRR-only OISST analysis This is updated thorough out the manuscript

Line 7: SST is not from model, it's a parametrization The following sentence has been edited " (for ice concentration and SST this is an estimate from SST parametrization)"

Line 11: "model estimate" should be "background model estimate" (as it's the background error in data assimilation terms) Changed

Line 13: As sigma_o is different for sea ice concentration and SST and described below, remove from this line. Also "parameter may vary spatially" is confusing without additional explanation.

Line 15: If above 65% is 10%, this should be > 0.10 based on your stats given on page3, line 16. The following sentence has been edited to clarify "ice concentration error depends on various atmospheric factors for ice concentration values less than 65%."

Line 16: I think this is intending to say something like "ensure that the assimilation is heavily weighted to the model background when there is large variation between the model and the observation." Needs rewording as it's unclear. However, method will weight towards model background even if the observation error is similar to the background. Why? The following text had been edited for the clarification "ensure that the assimilation is heavily weighted towards the observation when there is large variation between the model and the observation."

Line 22-25: Needs rewording as it's unclear what this means, and how this mechanism might directly affect the results. This is by intuition if there is a new ice area added to the cell that means some thickness exists in the cell otherwise what is the point in increasing ice area. If ice area is removed then the ice thickness has also to be removed from the cell.

Page 7 Lines 1-2: Reword this as implies model assimilates SST instead of ice in data gaps. Also gap between AMSR-E and AMSR2 should be mentioned here. Need to state that the model is free-running during periods where no data is available for assimilation. The following statement had been included to clarify the data gaps

"The AMSR-E instrument stopped producing data since October 2011, and AMSRE2 data has been used for assimilation beginning August 2012." "e.g. from 24 March 2005 to 31 March 2005 AMSR-E data are not available and, in that case, 'M2' assimilates SST instead of ice in data gaps. The AMSR-E instrument stopped producing data since October 2011, and AMSRE2 data has been used for assimilation beginning August 2012."

Line 4: "error" should be "mean difference", as the dataset being used as a reference is not necessarily "truth". This needs to be changed throughout the paper. Here, this should also say "absolute mean difference of ice concentration" for clarification. "OSI SAF" should be "OSI SAF dataset" (or similar wording). The corrections are made as follows "column 1 shows the absolute mean difference of ice concentration between the non-assimilated model and the OSI SAF data, column 2 shows the absolute mean difference of ice concentration of the model assimilated only with ice concentration and OSI SAF data, and column 3 shows absolute mean difference of ice concentration of the model assimilated with both ice concentration and SST and OSI SAF data. Model M2 shows improvement in the ice concentration for January and March, but the results do not improve significantly"

Line 7: "the results do not improve much" Is this compared to Model M1? But in some

locations the difference has reduced by about 20%, which is a good improvement. However, as you are assimilating the AVHRR-only OISST analysis, it is important to note that the product makes use of SSM/I and SSMIS ice concentration data to determine SST at high latitudes (though probably a different algorithm to the OSI SAF product). This means the SST observations you are assimilating are not truly independent from the SSM/I and SSMIS data you are using for validation. However, the AMSR-E/AMSR2 data is independent from the SSM/I and SSMIS data, and this should be stated. As stated previously the assimilation is heavily weighted towards observations in the ice edges. Also, AVHRR SST is inferred from ice concentration only when ice concentration is greater than 0.5.

Page 8 Figure 3: Need to state that this is ice concentration and which product the models are being compared to in the figure caption. It also needs to be stated in the text somewhere what the spin-up period of the model is. The following text has been included "The absolute mean difference of ice concentration for models M0, M1 and M2 from January 2010 to September 2011 is shown in row 1 and from August 2012 to December 2015 is shown in row 2" "The model was spin up for 3 months before assimilation, since no coupling with ocean model is done, the spinup time of 3 months is enough to estimate the ice conditions."

Lines 2-3: Need to state which instruments the assimilated ice concentration and OSI SAF data use. SSMIS. Added in text

Line 4: Why only giving the 2010 results? Also broken down into seasonal results would give a better picture. This was provided as an example to the reader. We wanted to understand the model bias as this would enable with tuning the model further. Seasonal results would be averaged and the bias may be reduced further. good

Line 8: This last sentence does not relate to anything shown in figure 4, remove this or improve discussion. Removed the sentence.

Page 9 Line 2: Which model thickness category are you using for the comparison?

[Figure]

The following sentence clarifies the point "For comparison and validation, ice thickness data from both the model and observation where the observed ice thickness has an uncertainty less than or equal to 100 cm are selected."

Line 2: observations from which instrument? Clarified in the following statement " The large unacceptable uncertainties in observation data derived from SMOS create difficulties for the analysis."

Lines 2-3: Unacceptable uncertainties in all observations? Confusingly worded. Clarified in the following statement " The large unacceptable uncertainties in observation data derived from SMOS create difficulties for the analysis."

Line 4: An uncertainty of 100 cm seems a lot for thin ice. What maximum ice thickness from SMOS are you using? From figures 5,6,7 it looks like 60 cm but this needs to be stated and explained. E.g. Xie et al. (2016; The Cryosphere, 10, 2745-2761) only use SMOS observations of < 40 cm, but others use up to 1 m thickness. The following statement clarifies the point "it is strictly recommended not to use the SMOS data with an uncertainty greater than one meter (Tian-Kunze and Kaleschke, 2016) for practical applications. For comparison and validation, ice thickness data from both the model and observation where the observed ice thickness has an uncertainty less than or equal to 100 cm are selected."

Line 5: How is model uncertainty determined? Model uncertainty is not determined. The way the data was selected for comparison and model observations are stated in the following sentence "For comparison and validation, ice thickness data from both the model and observation where the observed ice thickness has an uncertainty less than or equal to 100 cm are selected. The SMOS thickness has less uncertainty for thinner ice and higher uncertainty for thicker ice"

Selecting data this way gives a much better view of where the model data lies when compared with the uncertainty limits of observation.

[Figure]

Line 9: Add "As ice thickness increases through the season, so do the uncertainty limits." Added the sentence

Line 9: MO and M1 are too, except February 2013. What real benefit is the assimilation giving? Bringing closer to observations, but still deviates and are all still too high. Not convincing that it is only in the uncertainty range as even the free-running model is also managing that most of the time. The assimilation bring model values close to the observation. It shows deviation during March and April. For example, if we consider January-February of 2013 the values are within the uncertainty limits with the assimilation.

Line 10: Add "from October" before "until the end of February". The following sentence has been modified "The values of Model M2 are within the uncertainty limits of SMOS ice thickness from October until the end of February (except for 2014) end."

Lines 10-11: Move discussion of uncertainties to previous paragraph. Moved.

Line 12: Remove sentence beginning "Compared with the uncertainty values..." as this repeats information already stated. Removed.

Lines 16-19: This is because the assimilation is strongly weighted to model background. Demonstrates this is not the optimum set-up. If changing the value of alpha is expected to improve matters, why has this not been done? Not because assimilation is strongly weighted to model background. Assimilation is weighted heavily towards observation along the ice edge. We were looking whether improving ice edge would lead to improved results. Also, assimilating ice concentration includes the update of state variables related to variable drag parametrization. A sensitivity study/optimization work is going on and will be soon published in a separate work.

Figures 5,6,7: Combine these into one figure. The correspondence with the observations is poor after about January. All modelled ice is too thick, and although results for the M2 model are closer to observations than the M0 or M1 models, results are

still not very good. However, for results where the model thickness < 30 cm (figures 9,10,11) the models seem to be underestimating ice thickness rather than overestimating. This difference needs to be addressed in the paper. The figures have large data gaps and hence for clarity was presented as three figures. Also, the model thickness < 30 cm will show an underestimation of ice thickness since the assumption of 100% ice concentration in the algorithm for estimation of ice thickness from SMOS.

Figure 7 caption: make clear that M1 is not assimilating ice concentration because there is no AMSR data available. Rephrased as "The ice thickness from models M0, M1(not assimilating ice concentration as there were no AMSRE data available, but used the initial conditions from the model assimilated with ice concentration), M2 (assimilated only with SST and used model initial conditions derived from assimilating both ice concentration and SST) and observations (SMOS ice thickness) from October 2011 to April 2012. The uncertainty of observation (SMOS ice thickness) is shaded in gray."

Page 11 Lines 2-3: Why does figure 8 include regions where observed uncertainties are larger than 1 m, when on page 9 you have stated that this data has been rejected? This makes the figure very difficult to interpret, as it implies the model is underestimating ice thickness, but the comparisons in figures 5,6,7 indicate it is actually overestimating ice thickness for thin ice where the SMOS observations are more reliable - or underestimating for figure 9,10,11. Need to redo figure 8 showing only the relevant data, and also include panels with M2 differences to SMOS. Please see that the explanation for the figure was provided in Page 11 as follows "The Model M2 thickness, SMOS derived ice thickness, and the uncertainty of the SMOS derived measurement for 15 December 2010, 15 January 2011 and 15 March 2011 is shown in Figure 8, and includes regions where observed uncertainties are larger than one meter."

Page 12 Figure 8 caption: only showing for 3 individual dates, not 2010-2011 - update caption to reflect this. Caption has been updated as follows "The model 'M2' estimated ice thickness, SMOS-MIRAS derived ice thickness, and the observation uncertainty for

15th December 2010, 15th January 2011 and 15th March 2011."

Page 13 Figures 9,10,11: Maximum model thickness looks like 20 cm rather than 30 cm. Model underestimates thickness from December as thickness remains roughly constant throughout the year after this date. Also the assimilation makes little difference. Reasons for these results need to be addressed in the paper. Also caption states only model M2 is shown, but all models are shown on plot, update caption. Maximum model thickness looks like 20 cm since we considered only thin ice categories (< 30 cm) from the model. The results are averaged over the domain. Also, the observation has to be compared with the uncertainty of the thickness.

Page 14 Figure 11: What is the cause of the discontinuity in SMOS ice thickness and uncertainty between December and January? This needs to be addressed in the paper. We think it was due to the process of product generation.

Figure 12: Needs to be larger, as it is difficult to see the shaded regions. Made Figure 12 larger to fit the page width

Line 1: How is the "observation uncertainty" generated? Is this actually the AVHRR-only OISST analysis uncertainty? Add this to text. This is not independent data as it's being assimilated for model M2. Could choose a different dataset for validation. Yes. This is AVHRR only OISST analysis uncertainty. We are validating with the same data since we do not have access to any other reliable SST products.

Line 2: Sentence beginning "The SST assimilation..." does not refer to figure 12. It is confusing to have this sentence here with no context Rephrased as "In general, the SST from AVHRR-only OISST assimilation improves the ice concentration and ice thickness results for the model M2."

Line 3: The model doesn't have "outliers", results show it has systematic biases in both summer and winter. Rephrased as The assimilated model M2 still has systematic bias during the summer and winter, which may be improved by decreasing choice of

$\alpha$ (=6, presently) and by decreasing the nudging time scale (presently for SST nudging scale is 30 days)."

Lines 3-11: These lines give speculation on how the results could be improved, but this work needs to be done. The work on optimization and sensitivity analysis is still an ongoing work and will be published soon. Several other factors that affects the update of state variables (e.g. ridged ice area, melt ponds and albedos) in the equations of Tsamados et al have to be examined,.

Page 15 Line 2: Need to describe the method here, as Prasad et al. (2016) is a non-peer reviewed conference paper. Please note that the methodology is also described in "section 3. Remote Sensing Data for Assimilation and Validation" The following sentence had been included for clarity "The ULS measurements were separated into level ice draft and keel depth measurement as described in Prasad et al, 2016 and also in Section 3."

Line 6: Add that rho_w is the density of water. The following sentence has been included to clarify " w = 1026 kg/m^3 is the density of sea water"

Line 7: "about 10 cm" - give specific value (variation with season? Different years?). Need to add RMS or standard deviation. This is with respect to whole year also it is mentioned that this is absolute error.

Line 8: An error of 10 cm on a draft of 20 cm is proportionally very large, so can't be described as good correspondence. Changed: "The error of 10cm on draft of 20 cm can be accepted considering large difference in spatial resolution between the ULS and Model." Line 8: Reiterate here for benefit of reader that only done analysis for 2005, 2007, 2009 as this was when data was available. The following sentence was included for clarification " Also, the analysis was done only for 2005, 2007, 2009 as this was when data was available"

Line 10: "close to the location of the ULS" - are you interpolating the model result to

the observation location? If not what method is being used for the matchups and why?

The following sentence clarifies it. The point that is close to the location of the ULS is used for comparison. The model is also an averaged daily values.

"The discrepancy occurs due to the fact that ULS gives values at a particular location with high resolution (within the footprint of several meters), while the 10 model is of 10 km resolution gives an averaged result close to the location of the ULS."

Page 16 Lines 1-2: "single melt pond" - even in winter? This method needs more description Please note that this has been described in Tsamados et al, 2014 and a reference to the same has been provided.

Line 5: $H_k$ is not used in equation (4), remove (given below for equation (5)). Removed.

Line 6: $m_r$ and $m_k$ need more explanation - slopes given in degrees but what are 0.4 and 0.5? Clarified in the following sentence "$m_r = \tan(alpha\_r) = 0.4$, alpha_r = 21.8 is the slope of the sail and $m_k = \tan(alpha\_k) = 0.5$, alpha_k = 26.5 is the slope of the keel"

Lines 5-8: Where are these values obtained from? Not all the variables have been given values either. These are values are from (Shokr and Sinha, 2015) given references and some are default values in CICE

Line 12: Citation required for this statement. Citations are provided (Peterson et al., 2013)

Line 13: Model and observation of keel depth or ridge height? Confusing. Rephrased as "Here the model and the observation of keel depth are used to estimate the parameter C."

Line 16: Figure 14 only shows modelled and observed keel depth, not ridge height so can't see this relationship. Also need to give statistics for difference between modelled

and observed keel depth. This part only does an estimation of parameter C form the keel depth of the model and the observation.

Lines 17-19: If further work may result in a different conclusion, you need to do this further work to be able to draw any conclusions. This requires tuning of model and assimilation paramters and is an ongoing work.

Line 25: How did you calculate lead fraction? Or cite existing product if that is what you used. Citation made "The presence of leads was ensured by selecting the regions where lead fraction derived from CryoSat-2 (Ricker et al. 2014) was greater than zero."

Line 26: Need to clarify in the text that the uncertainty given is for CS2 freeboard measurements. Need more information on the CS2 (CryoSat-2) product, e.g. how often available, where data was accessed etc. The information is provided. The product citation is provided and also mentioned in Section 3 "For the region, the uncertainty of the freeboard measurements is below 40 cm (Ricker et al. 2014) "

Equation (6): I can't find this equation in Tsamados et al. (2014) but it looks like it's missing some brackets This is only a shortened version of the equation (26) in Tsamados et al. 2014, the equation is represented in terms of draft. If the draft in the equation(6) is replaced by equation (3) and working out some math the same equation (26) in Tasamdos et al., 2014 can be derived.

Page 17 Line 1: "absolute difference" should be "absolute mean difference". Changed

Line 2: "M2 freeboard measurements are close to the observed freeboard". This isn't true - figure 15 shows that the differences between the model runs and the observations are a large percentage of the actual values. There is also variation between the different months shown. It would be better to show differences rather than absolute differences on the spatial plot to be able to see where the biases are and in which direction. Other statistics such as RMS or standard deviation also need to be given

Our main intention was to see if the model and observation dereference is higher than

the observed uncertainty which is 40cm for the region of interest. We will be doing further statistical analysis to quantify the model error. Moreover, the observation has large uncertainty which still leaves a doubt of computing RMSE would give much sense. Also, please see Figure 17.

Page 19 Figure 16: Caption should specify Jan, Feb, March 2011 and not just 2011. Also need to show the difference plot and give other statistics e.g. RMS. Clarifications had been made to captions.

Line 4 (and line 6): The model values look systematically different to the observations in Figure 16. Figure 17 shows that the model is unable to replicate the seasonal changes in the freeboard observations, and just increases throughout the year.

Line 5: The data presumably still undergoes averaging if the points are observed multiple times within that month. Much more information on the dataset is needed. Freeboard equations are dependent on the ice volume and ice concentration and hence the systematic errors observed in thickness and concentration will affect the freeboard too. Also, please note that the freeboard measurements are a monthly mosaic of the data collected (yes, some averaging will be going on) and also, has 40 cm of uncertainty. It is by comparing with the observation uncertainty we are saying the model values are within the range of observed values

Page 20 Lines 2-3: Needs references to back this up. Added references (Lemieux et al., 2016, Rae et al., 2015)

Line 6: Misleading, as have not validated the assimilation method itself, only assimilated different combinations of observations Lines 6-7: This sentence implies the model is assimilating all these variables, which is incorrect. Reword

The sentence has been rephrased as "The results from the updated model were compared with satellite derived measurements to validate the model estimates of ice concentration, ice thickness, freeboard. Moreover, the model results were used to estimate

relationship between sail and keel depth."

Line 8: Disagree that it is a good correspondence. This statement is made except for the maximum ice extent when the ice thickness goes beyond the uncertainty limits of the observed thickness.

Line 10: The RMSE should be mentioned previously with the rest of the results. RMSE is mentioned in text.

Line 11: Where have you split results into below 40 cm? Seems to be 60 cm for observations or 30 cm for model category (which looks more like 20 cm). Sorry for the confusion. Here describing about the freeboard not ice thickness. Please see that the paragraph has been rearranged.

Lines 14-15: No remote-sensing ice thickness data is available in the summertime due to the presence of melt ponds. As summertime data has been excluded in this study the error contribution from melt ponds is likely to be small. The meltpond evolution of the model will be a future part of the study.

Line 18: Agreement is not always close. This statement needs to be revised. Revised

Lines 20-21: This needs to be done, rather than stating it as future work. Modified

References: Many of the references are missing important information, for example access URLs for technical reports, and format type e.g. book, report, dataset etc. Technical corrections There are a number of instances of citations not being in brackets when they should be, and vice versa. The authors need to go through the manuscript carefully to correct these. The authors need to ensure that the paper is read by a native English speaker as there are a large number of minor but important grammatical errors. There are too many to list here. There are also several areas where information is repeated within the same paragraph. Some are listed above. Correcting these instances would improve the readability of the paper. References were updated.

---

## Author Comment (AC3) · 10 Oct 2018

Please note that the comments from referee are given in black font "Times new roman". Comments from the authors are given in blue font "Calibri". The changes in the manuscript are given in green font "Calibri".

General Comments

In this paper, the authors assimilate ice concentration and AVHRR-derived SST into a 10 km CICE model for Baffin Bay and the Labrador Sea for the period spanning from 2010-2015. A series of 3 experiments are performed to assess the model's performance against ice thickness from SMOS, ice draft and keel depth from a ULS, and freeboard estimates versus CryoSat-2. A control run does not have any data assimilation, while the other two assimilate SST and SST and ice concentration. A nudging and optimal interpolation technique based on Lindsay and Zhang (2006) is used. Model mean ice thickness is compared against the SMOS ice thickness for the periods of Oct – March for the years 2010-2015. Overall, the "M2" test case which assimilates SST and ice concentration performs best, and is generally within the uncertainty bounds of the SMOS data; however there is a significant positive bias shown for all years. An impressive comparison of the model's (M2) keel depth versus a ULS for 2005, 2007

and 2009 show very good agreement with data. However, model freeboard differences with CryoSat-2 data for Jan, Feb, and Mar 2011 show very little difference amongst the three test cases. Overall, while not "state-of-the-art", this paper shows some improvement with the assimilation of SST and ice concentration in a regional ice modeling system. I recommend publication with minor revisions.

We would like to acknowledge the reviewer for the comments and suggestions.

**Specific Comments**

How are ice boundary conditions addressed in the model? Same technique as discussed in Prasad et al. 2015 paper? If yes, state this in the paper.

The following text has been included for clarification

"The net heat flux from the atmosphere is the upper boundary condition for ice thermodynamics. The heat flux from the ocean to the ice is the lower boundary condition. Based on temperature profile and boundary conditions the melt and growth of ice is computed. The open boundaries are configured in the same way as in (Hunke et al., 2010, Prasad et al., 2015)"

You use a 35-50 km SMOS ice thickness product for your thickness comparisons. You state that the SMOS data should not be used for thickness greater than 1 m; Figure 8 (middle column) shows a significant area of ice thicker than 1 m by March 15, 2011.

Why didn't you consider using a merged CryoSat-2/SMOS ice thickness product such as is available from AWI? Do you have plans to assimilate ice thickness or freeboard into your model?

Figure 8 has been described in the following sentence

"The Model M2 thickness, SMOS derived ice thickness, and the uncertainty of the SMOS derived measurement for 15 December 2010, 15 January 2011 and 15 March 2011 are shown in Figure 8, and include regions where observed uncertainties are larger than one meter"

During the time the merged product was not available. We will use the merged product in the future study. Yes, we do have plans to combine other products for assimilation.

Page 1 line 19: why limit discussion to "climate forecast researchers"? This is important for operational sea ice modeling as well.

The following text has been modified

"The climate forecast researchers and operational ice modeling communities depend on numerical modeling techniques implementing the physical process of atmosphere and ocean on large scale computational platforms along with data assimilation methods to retrieve the information on sea ice parameters."

Page 3 lines 7-8: Why does the assimilation begin in January 2005? If the model is started from a no-ice state in September 2004, why doesn't assimilation begin in October 2004, when you should have data?

Please note that the AMSRE ice concentration product was available from January 2005 and hence assimilation started from the same period. Also, the model was given a 4 months spin-up.

Page 3 line 11: Explain how you use AMSRE for validation of the model if you are assimilating that same data?

This was corrected AMSRE was used for assimilation and the product was compared with OSI SAF data.

" Ice concentration derived from AMSRE of resolution 6 X 4 km (Spreen et al, 2008) were used for the assimilation of ice concentration."

Page 4 line 1: What do you mean by "erroneous data"?

The following text has been modified for clarification.

The erroneous data, were the ice concentration error was 100% or retrieval algorithm has failed were filtered out before the comparison.

Page 7 lines 1-2: Why does M2 only assimilate SST when there are gaps in AMSR-E (and I assume AMSR2)? Why not assimilate all the time?

M2 assimilated SST only when ice concentration is not available for assimilation, otherwise the model assimilated both SST and ice concentration.

Page 8: Why is there no discussion on error reduction for the period from Nov 2012 – Dec 2015? A table of error stats would be helpful here.

Only an example has been provided here. The rest of the results are shown as Figure 4.

Page 9: Have you tested different values of α?

Yes, different values of alpha were tested. A sensitivity of the parameter alpha has been shown in Lidsay et al. The value has to be further optimized considering the variable drag formulation variables for the model, which would be a future work.

Page 15: How is snow measured or estimated in the ULS data? I assume the model results shown in Fig. 13 are for M2? If yes, state in figure caption. How do M0 and M1 compare here?

Upward looking sonar measures the draft from below and the measurement of snow is not available. Since we were interested in the results of assimilated model only M2 results are given.

Page 17: I see little difference in Fig. 15 between column 2 and 3 in the plots. The paper states "M2 freeboard measurements are close to observed freeboard". I disagree. Perhaps the Jan 2011 looks best, but overall, the differences seem small for all 3 test cases.

Yes, these differences are very small. But M2 is found to be the best match with the observation.

Page 1 line 18: add "it" after "makes" and before "practically"

Included

Page 2 line 7: rephrase to "into CDOM using a 3D"

Rephrased

Page 2 line 9: replace with "Lindsay and Zhang (2016)"

Replaced

Page 2 line 16: "extent were overestimated"

Changed

Page 2 line 19: "of the CICE model"; which version of CICE is used? Specify in text

Rephrased as

"CICE version 5.1.2"

Page 2 line 20: ", and the combination"

Rephrased

Page 2 line 21: "(Lindsay and Zhang, 2006; Wang et al., 2013)"

Changed

Page 2 line 22: replace "cheap" with "inexpensive"

Replaced

Page 2 line 24: "Baffin Bay and the Labrador Sea". resolution. . ." "This work uses a high-resolution..."

Changed

Page 2 line 33: replace with "Density-based criteria (Prasad et al., 2015) to compute. . ."

Rephrased as

" Density-based criteria were used as in (Prasad et al., 2015) to compute the mixed-layer depth and thereby compute the SST and the potential to grow or melt sea ice."

Page 2 line 34: replace "fo" with "of"

Replaced

Page 3 line 3: give reference for and spell out NARR

References provided and NARR is replaced with North American Regional Reanalysis

Page 3 lines 7-8: reword last sentence as it is confusing. Page 3 line 10: "remote sensing data sets"

Rephrased as

"The data assimilation starts from January 2005 and continually assimilated whenever data was available."

Page 3 Table 1: add a column with dates for AMSR2

Added

Page 3 last line: "The OSI SAF product"

Added

Page 4 line 2-3: reword sentence to "Measurements derived from AVHRR (Reynolds et al., 2007; Smith, 2016) were used for SST assimilation."

Reworded

Page 4 line 18: "ice thickness (observations) shown in Table 2 include. . ."

Rephrased

Page 4 line 19: "knowledge of the"

Corrected

Page 4 line 22: "cover and the onset"

Corrected

Page 4 line 25: "from an ULS"

Corrected

Page 6: Reword first 2 sentences as they are poorly written

Rephrased as

"The assimilation module uses a combined optimal interpolation and nudging technique for ice concentration (Lindsay et al., 2006, Wang et al., 2013). The method can be represented generally as equation (1) (Deutch, 1965, lindsay et al., 2006)."

Page 5 line 12: here and in numerous places in the text, use "Lindsay and Zhang (2006) " as the reference, not the way it is written in the paper.

Corrected

Page 7 lines 1-2: add parentheses to "AMSR-E (e.g., from 24 March 2005 to 31 March 2005), AMSR-E. . ."

Included parenthesis

Page 7 line 7: reword to "little improvement between M1 and M2 is shown for May 2010.

Rephrased as

" Model M2 shows improvement in the ice concentration for January and March, but little improvement between M1 and M2 for May 2010. "

Figure 2: Makes dates Jan 2010, Mar 2010 and May 2010 BOLD black so they are easier to see. Caption is labeled wrong as it should be for Jan 2010, March 2010, and May 2010.

Corrected

Page 3 Figure 3 legend: reword to "The absolute error for models M0, M1, and M2 from a) January 2010 to September 2011, and b) August 2012 to December 2015.

The caption has been reworded to "the absolute difference for models … "

Figures 5-7: Make the uncertainty shading a lighter gray as it is difficult to see the otherlines. What is the bias and RMSE for these comparisons?

Here we consider whether the model values are within the uncertainty limits of the observation.

Figure 8: same comment about making dates on plots readable. Can you add another column showing the difference between M2 and SMOS-MIRAS?

The figures has to be read with the uncertainty limits of the observation. Moreover, from Figure 5, 7 the uncertainty goes higher during late winter.

Figures 9-11: Make grey uncertainty lighter for readability. Captions not correct as M0 and M1 are shown (not just M2) as stated in Figure captions.

Captions are corrected.

Page 14: Explain the difference in uncertainty shown in Fig 11 from Dec 31 to Jan 1.

The following line has been added for clarification

" the shaded region shows the uncertainty of the thin ice from SMOS data"

Page 14 line 7: reword to "blow-ups" or something similar
Reworded

Page 19: Make dates legible on all 9 plots.

Made clear

Page 20 line 13: sentence with "November end but lies" does not make sense.

Changed to

"The thin ice category thicknesses are overestimated from October to November end but the values are within the uncertainty limits of SMOS from December to March."

Page 22: provide a more complete reference for Deutch 1965

Provided

Page 23 line 21: Provide a date for first Tietsche reference

Date provided

---

## Author Response (AR2)

**Author Response to Anonymous Referee 1**

The authors would like to acknowledge Referee 1 for the comments, which helped improve the manuscript.

I previously provided a detailed review of the initial submission of this paper. Unfortunately, in their resubmission, the authors have not adequately addressed serious issues with the work. Below I have given feedback on the authors' responses to the main comments from the previous review. Since the main points have not been well addressed I have not gone through the responses to the detailed specific comments.

From initial review: "(1) It is unclear what is novel about this work. The conclusion states that the authors' use of a variable drag formulation is unique. However, Tsamados et al. (2014) previously incorporated a variable atmospheric and oceanic form drag into the CICE model. The paper has been cited by the authors, but they do not describe how their implementation of this method is related to any of their results. At the most basic level, the authors should show results with and without this formulation.

The statement that other centres do not provide details of parameters other than ice concentration or thickness is inaccurate, particularly as many of those centres are also using the same CICE model as the authors, with the same available parameters. The main conclusion of the results seems to be that assimilating observations brings the model output closer to reference observations, which is not a new result. Perhaps the differences in results when using thin ice of < 60 cm compared to < 30 cm might be an interesting angle, but this is not explored."

Authors' response: "The authors would like to thank the reviewer for the comments and suggestions.
It is true Tsamados et al. (2014) has incorporated a variable atmospheric and oceanic form drag into the CICE model, but have not been used with a data assimilated model that produced parameters such as freeboard, sail, keel measurements. The forecast centers are using CICE model a constant variable drag formulation. Moreover, the used models are coupled with ocean models."

New review comment: "This does not adequately address the comment that the authors have not demonstrated that any of their results are related to the variable atmospheric and oceanic form drag. Other comments have not been addressed."

The novelty of the manuscript includes a regional implementation of CICE model with data assimilation and validation of CICE model with satellite and in situ (ULS) observations that was never performed before. Considering a wide usage of CICE model the results can be interesting for the readers.

Tsamados et. Al (2014) has already performed the variable drag formulation, however the work didn't use data assimilated model. We were interested in including the variable drag formulation for the assimilated model. The use of variable drag formulation helps estimate the freeboard, level ice draft, ridge height and keel depth. Without the usage of variable drag formulation these parameters cannot be estimated.

From initial review: "(2) The statements throughout the paper, that the model fits the validation observations well, are not backed up by the results themselves. Although the assimilation improves results, there remain clear systematic differences between the model output and the reference observations."

Authors' response: "The study estimates the bias observed over years and will use it for further tuning of the model."

New review comment: "This does not address the statements in the paper that the model fits the validation observations well."
Please note that the authors admit that there is a variability during the winter period.

From initial review: "(3) There are many omissions in explanation of methods, several contradictions in the text, confusing wording, and the paper is missing references to current literature and relevant similar systems, e.g. TOPAZ, RIPS/RIOPS, ACNFS. Also missing is a description of how the authors' system differs and improves on these, and indeed what the purpose of the new system is. A number of the citations given are conference papers or otherwise unpublished works, which are not peer-reviewed and should not form a significant basis of citations."

Authors' response: "The assimilation and tuning of the model is still an ongoing work and will be compared with RIPS and other models in the future. Moreover, the cited conference papers include the developments of RIPS. The novelty of the manuscript is the regional implementation of the uncoupled model."

New review comment: "This particular comment was not a suggestion to quantitatively compare results with other systems, but to describe the purpose of the authors' system and how it fits in with (and enhances) current work."

As mentioned earlier RIPs still uses a constant drag formulation in their assimilated model. Here we use a variable drag formulation for the assimilated model, which produces estimates of freeboard, keel depth and level ice draft.

General comments

From initial review: "The relevance of how this work fits in with the published literature needs to be discussed, along with other regional modelling systems. How do the results compare to e.g. coupled ocean-ice systems?"

Authors' response: "The coupled ice-ocean systems are more complex and require additional tuning of ocean parameters. The purpose of the study was to go without an ocean model using a mixed layer parametrization and data assimilation to produce the best results. Coupling would require extensive work and is out of the scope of the current work."

New review comment: "This was not a suggestion to do this work, but to compare your results with those already available in the literature."

The regional model with sea ice component was implemented for Baltic Sea with closed boundaries, 3DCEMS. Here we are simulating the conditions for Baffin Bay and the Labrador Sea. Additional reference was added with the following text:

"The 3D-CEMBS is an eco-hydrodynamic model that includes a coupled POP-CICE model for operational forecasting implementation of CICE on a regional scale. The implementation on the regional scale of the ice component and the validation work is still ongoing (Dzierzbicka-Głowacka et al., 2013)."

From initial review: "What is the benefit of only using an SST parametrization? Is this system to be used for operational or research purposes? A large number of the references used in this paper are unpublished or non peer-reviewed works including conference papers. A more complete discussion of the peer-reviewed literature is necessary."

Authors' response: "Please note that the SST parametrization was discussed earlier in Parasad et al. (2015). Similar to RIPS 2016 the model used the same density parametrization but with a different criteria for forwarding the slab ocean model parametrization."

New review comment: "This does not address the need to include a more complete discussion of the peer-reviewed literature, or to add information on the purpose of the system."

"Please note that the SST parametrization was discussed earlier in Parasad et al. (2015). The benefit being a standalone and less complex ice model. Similarly RIPS 2016 the model used the same density parametrization but with a different criteria for forwarding the slab ocean model parametrization."

The following text has been included to clarify:

"The ice prediction models such as Regional Ice Prediction System (RIPS) (Lemieux et al., 2016) limits the discussion on ice concentration estimates from the model. In this work, in addition to validation of the ice concentration we also discuss the effect of the assimilation on ice thickness, freeboard, draft and keel depth."

"(Prasad et al., 2015) used a density criteria of 0.2 Kg/m 3 at 10m depth, the other models such as RIPS by CIS (Lemieux et al., 2016) uses a density criteria of 0.01 Kg/m 3 from the ocean surface."

From initial review: "The paper needs more information on all the input and validation data sets, including descriptions and data access information. The authors also need to ensure that all the datasets used are properly cited."

Authors' response: "All the input and validation data sets had been described and properly cited."

New review comment: "Do you mean this has now been updated? This was not in the first draft

of the paper."
This was updated in the second draft.

From initial review: "Why is the assimilation system set up to weight heavily in favour of the model rather than giving equal weight to the observations?"

Authors' response: "The assimilation followed work of Lindsay et al. Further tuning of the model physics and assimilation schemes have to be done to better understand the model behavior. The system is actually weighted heavily in marginal ice zones where significant bias can occur."

New review comment: "This doesn't actually answer the question on why it was set up this way."
Please note that chances of significant bias occurs in the marginal ice zone/ice edge and hence the system was "weighted heavily towards observation" in marginal ice zones.

From initial review: "The paper repeatedly states that AVHRR data was assimilated. Actually, it looks like the authors are assimilating the AVHRR-only OISST analysis product, which although based on observations, is an analysis product. This needs to be made clear, along with information on the temporal resolution, timeliness etc of the product. Additionally, this product uses SSM/I and SSMIS information to create proxy SST observations for assimilation at high latitudes. This means the SST observations also include input from ice concentration data. Therefore, they are not independent from the SSM/I and SSMIS data being used for validation."

Authors' response: "Yes, SST used an analysis product since model is assimilated over its domain including at ice edges where ice concentration has lower values. It was clarified in the paper that AVHRR-only OISST analysis product uses ice information to retrieve SST only for regions where ice concentration is greater than 0.5. The following sentence has been included for clarity " The analysis product estimates SST from ice concentration only in regions where ice concentration is greater than 50%, otherwise uses satellite data to retrieve SST values."

New review comment: "This additional sentence only partly addresses the points raised in the comments."
Please note that we have already demonstrated a comparison of model that is assimilated only with ice concentration, which improves the model results, the SST assimilation only makes it better. The results without SST assimilation but with ice concentration assimilation were already shown.

From initial review: "More detail and justification of which thickness ranges of SMOS and CryoSat-2 observations are being used is needed."

Authors' response: "The justification has been provided in the paper "Also, it is strictly recommended not to use the SMOS data with an uncertainty greater than one meter (Tian-Kunze and Kaleschke, 2016) for practical applications." This has been clearly stated in the user manual of the SMOS product. For Cryostat-2 freeboard measures had been used for comparison. Since thickness estimates of CryoSat 2 is derived from freeboard estimates we

think that it would be best to compare freeboard estimates with Cryosat-2 instead of thickness estimates."

New review comment: "Data provided by CryoSat-2 and SMOS have very different characteristics. Using the uncertainty to reject SMOS data is a reasonable method, but what range of thickness data does that mean you are using? What about CryoSat-2 data? The point still stands even if you are using freeboard rather than thickness. There is a lot of information available in the literature. Different users make different decisions on what data to accept or reject, but justification of the decisions is necessary."

Yes, we understand that the data provided by CryoSat-2 and SMOS have very different characteristics. Here the thickness was compared with SMOS product where uncertainty of the thickness was below 1m, which was recommended for all applications. If such thickness is selected, the thickness range can vary depending on the factors that affect it given in Table 2, and hence instead of mentioning thickness range uncertainty was used to select it. Moreover, freeboard measurements of the model were compared with the CryoSat-2 freeboard measurements. Please note that CryoSat-2 does not measure thickness but only the freeboard.

From initial review: "What real benefit is the assimilation giving? Figures 5-7 show that it brings the model closer to the observations, but it still deviates and all modelled ice thickness is too high. It is not convincing to state that the M2 model has good correspondence with the observations due to being in the uncertainty range, as even the free-running model is also managing that most of the time. Assimilating SST in addition to sea ice concentration produces better results, but few if any operational centres will not already be assimilating ice concentration and not SST observations."

Authors' response: "All modeling centres assimilate SST in Ocean model and pass the information to sea ice model. Here, we state that if we use analysis product of SST in the ice model itself it produces better results."

New review comment: "In order to make that statement, a quantitative comparison needs to be given in the paper. No evidence for this is provided."

As mentioned, we have compared the model with the observation based on the uncertainty range of the observation data. In addition, absolute error was provided for the comparison purpose.

Authors' response: "Moreover, operational centers give information on ice concentration but ice thickness significant biases are observed also, other information such as ridge height, keel depth, freeboard data are rarely discussed."

New review comment: "Are you planning to make this information available operationally? If so, should be stated in the paper."

We are planning to make this operational after sensitivity studies.
Added to the text.

From initial review: "The authors acknowledge that the assimilation is not optimized. If changing the value of alpha or adjusting the nudging timescale is expected to improve results, why has this not been done? Similarly for the relationship between ridge and keel."

Authors' response: "This is a parameter sensitivity study and is an ongoing work."

New review comment: "If key work is still ongoing, it needs to be considered if the work is in a position for publication."

No author response given to any of the following comments from initial review: "The authors need to also include RMS (or standard deviation) statistics, as well as mean difference when discussing how well models match validation observations.

Please note that the RMS and the anomaly were discussed in Prasad et al, 2015 and this manuscript is the continuation. To give the reader an idea of RMS related to ice thickness, it was provided using freeboard.

For figures 5,6,7 the modelled ice is too thick for all model runs after January. Although results for the M2 model are closer to observations than the M0 or M1 models, results are still not very good, which is not mentioned in the paper. In general, throughout the paper, systematic biases or errors which are large in proportion to the model variable values are not addressed, or are dismissed as being within uncertainty levels. This shows a poor understanding of the validation results.

For results where the model thickness < 30 cm (figures 9,10,11), the models seem to be underestimating ice thickness rather than overestimating. This difference to the results seen in figures 5,6,7 needs to be addressed in the paper.

For figures 9,10,11 the modelled thin ice thickness remains roughly constant from December, and also the assimilation makes little difference. Reasons for this need to be addressed in the paper."

The authors acknowledge that there are systematic biases observed in the model. This is partially due to the assumptions of no mixed layer heat flux, constant salinity.

[revised manuscript text omitted]